



# LPJmL-Med – Modelling the dynamics of the land-sea nutrient transfer over the Mediterranean region–version 1: Model description and evaluation

Mohamed Ayache[1,2], Alberte Bondeau[2], Rémi Pagès[1], Nicolas Barrier[3], Sebastian Ostberg[4], and Melika Baklouti[1]

[1]Aix Marseille Univ., Université de Toulon, CNRS, IRD, MIO UM 110 , 13288, Marseille, France
[2]Institut Méditerranéen de Biodiversité et d'Ecologie marine et continentale, Aix-Marseille Université, Technopôle Arbois-Méditerranée, 13545 Aix-en-Provence, France
[3]MARBEC, CNRS, Ifremer, IRD, Université de Montpellier, Sète, France
[4]Potsdam Institute for Climate Impact Research, Research Domain 1: Earth System Analysis, Telegraphenberg A62, 14473 Potsdam, Germany

**Correspondence:** Mohamed Ayache (mohamed.ayache@mio.osupytheas.fr)

**Abstract.**

Land forcing (water discharge, and nutrient loads) is reported as one of the major sources of uncertainty limiting the capacity of marine biogeochemical models. Runoff from rivers and coastal plains delivers significant amounts of nutrients to the Mediterranean Sea from agricultural activities and urban waste water. Several recent studies show that variations in river inputs

may play a significant role in marine biogeochemical cycles and the planktonic food web throughout the entire basin. The aim of this study is to estimate the water discharge as well as nitrate ($NO_3$) and phosphate ($PO_4$) release into the Mediterranean Sea from basin-wide agriculture and inhabited areas through the implementation of the biogeochemical land-sea nutrient transfer processes within the agro-ecosystem model Lund Potsdam Jena managed Land for the Mediterranean (LPJmL-Med). The representation of the nutrient transfer from land to sea has been introduced into LPJmL-Med by considering the following pro-

cesses: remineralization, adsorption, nitrification, denitrification and phytoplankton dynamics. A compilation of a new input data set of fertilizer, manure and wastewater nutrient content [1961-2005] has been added to the LPJmL-Med forcing data set.

The first basin-wide LPJmL simulation at 1/12° indicates that the model succeeds in simulating the interannual variability of water discharge for the main rivers in the Mediterranean Sea, especially the Po, Rhone and Ebro Rivers. A very high correlation (R-square values higher than 0.94) is found for these three rivers. Results also show a good consistency between the simulated

nutrients concentration ($NO_3$ and $PO_4$) and available in-situ data. River outflows of $NO_3$ and $PO_4$ exhibit opposite trends in the Mediterranean Sea. $NO_3$ showed a more or less continuous increase from the beginning of the 1960s until the present in all three rivers. $PO_4$ trends are more heterogeneous. There is a strong increase in $PO_4$ between 1960 and 1980, followed by a decrease in mean annual fluxes from the second half of the 1980s as a consequence of the banning of phosphates in detergents, and the construction of waste water treatment plants in the different countries. Results show that wastewater strongly contributes to the

river phosphate fluxes, while both agriculture and wastewater control the nitrogen (mainly as $NO_3$) fluxes from rivers to the Mediterranean Sea.



## 1 Introduction

Biogeochemical cycles are currently undergoing major changes due to human activities, such as intensive agriculture and fossil fuel burning (Gruber and Galloway, 2008; Bouwman et al., 2009; Fowler et al., 2013). The effects of anthropogenic

activities on biogeochemical cycles can be either direct (*e.g.* through an increased supply of nutrients to ecosystems), or indirect (*e.g.* through the rise in air and water temperature, which in turn alters the rate of key processes in nutrient cycling). This alteration of the biogeochemical cycles of key nutrients such as carbon, nitrogen and phosphorus may in turn heavily affect the whole structure and functioning of terrestrial and marine ecosystems. For example, the increasing use of nitrogen and phosphorus fertilizers for agricultural production can have devastating effects on both aquatic and marine ecosystems (*e.g.*

Diaz and Rosenberg, 2008; Scavia et al., 2014). The development of cities with increasing populations also affects the release of nutrients through wastewater (Morée et al., 2013), in turn affecting the functioning of aquatic and marine ecosystems. However, nutrient retention along the aquatic continuum can be substantial and have a buffer effect regarding the human-induced increase in nutrient supply to terrestrial and aquatic systems, although it does not balance nutrient release at global scale (Beusen et al., 2016; Lassaletta et al., 2012; Billen et al., 2007). Retention processes also modify the ratio at which nutrients are transported

downstream, as well as their chemical form, which can have important implications for biogeochemical cycles from local to global scale (Bouwman et al., 2013; Peñuelas et al., 2013). Assessing the amount of nutrients retained in soils and rivers is thus crucial to understand the impact of human-induced release of nutrients on aquatic and marine ecosystems.

Changes in agriculture in industrialized countries over the last 50 years have strongly affected the severity of the impact of anthropogenic activities on global biogeochemical cycles, mainly with regard to the nitrogen and phosphorus cycles (Bouwman

et al., 2009; Fowler et al., 2013). Switching from traditional to modern agriculture has resulted in a geographical separation of livestock and crop production, inducing an increased use of inorganic fertilizers to substitute for manure (Thieu et al., 2011). In areas with intensive livestock production, the content of nitrogen and phosphorus in soils has strongly increased, inducing major extensive leaching of these nutrients in soils and the contamination of freshwater (Thieu et al., 2011; Bouwman et al., 2013; Leip et al., 2015). In cropping systems, a large part of nutrients from manure and inorganic fertilizers is not assimilated by

plants and is thus transported through leaching and runoff along the aquatic continuum (Coskun et al., 2017). Eutrophication in coastal waters due to the increased inputs of nitrogen and phosphorus from groundwaters and rivers may indeed induce hypoxia or anoxia, and can lead to changes in the whole trophic web (Diaz and Rosenberg, 2008; Rabalais et al., 2010; Moller et al., 2014).

The Mediterranean region includes countries with a wide range of socio-economic and development backgrounds. A speci-

ficity of this region is the diversity of the agricultural practices in the different countries, characterized by differences in crop types, fertilization and irrigation methods, as well as livestock and manure quality (Rost et al., 2008; Potter et al., 2010; Fader et al., 2015). The Mediterranean region also includes a wide variety of rivers, which differ by their flow rate, their seasonality, their degree of exploitation for irrigation, and wastewater treatment policy. The region is subject to global change pressures such as increasing population and gross domestic product (GDP), urbanization, and climate change (Schröter et al., 2005;

Diffenbaugh and Giorgi, 2012; Zdruli, 2014). Its agricultural production will probably face higher vulnerability both with re-





gard to ensuring food security (several countries already requiring extensive food imports), and for exporting products that are essential to the national economies of many countries (Hervieu, 2006; Verner, 2012). Thus, it is important to understand how the effects of urbanization, socio-economic and agricultural development in Mediterranean landscapes translate into land-sea nutrient transfer dynamics.

The most important macro-nutrients for the marine ecosystem are nitrate and phosphate. Nitrogen (N) and phosphorous (P) occur in rivers in various forms, *i.e.* dissolved, particulate, organic or inorganic. Inorganic form is becoming increasingly important due to the growing anthropogenic contribution. Although N is generally considered to limit primary productivity in most of the world's oceans, previous studies have suggested that the Mediterranean Sea may be an exception, with a strong P limitation (Krom et al., 1991). The degree of P limitation increases from west to east across the entire basin (e.g. Krom et al.,

2003). For these reasons, and because of the greater availability of data on $NO_3$ and $PO_4$ as compared to the other nutrient forms, we focus hereafter on these two macro-nutrients. In-situ data are available only for the largest rivers (Rhone, Po, Ebro, but not Nile), because of their importance for the local economic development and geochemical budgets at large scale.

Different models simulating the transfer of nutrients from land to oceans have been developed over the last decades, with an emphasis on nitrogen and phosphorus cycles (*e.g.* Billen and Garnier, 1999; Seitzinger et al., 2005; Mayorga et al., 2010).

However, existing models simulating both nitrogen and phosphorus cycles along the aquatic continuum are not adapted to our case study as: (1) models with an annual time step do not represent the seasonal variability of river discharges, which is high in the watersheds flowing to the Mediterranean Sea; (2) complex models can be difficult to apply to the whole Mediterranean region, as it includes a wide variety of countries with strong irregularities in data availability. In this context, LPJmL is able to simulate in good detail the functioning of Mediterranean agriculture with a comprehensive representation of all vegetation

types (agricultural and natural) and the full terrestrial hydrological cycle in one consistent modeling platform (Fader et al., 2015). In this paper, we aim to present a new version of the Mediterranean LPJmL model (Fader et al., 2015), hereafter called LPJmL-Med, which simulates with a daily time-step the coupled carbon and water cycles of the actual ecosystems (natural potential vegetation and agriculture, including annual and perennial crops, and managed grassland) and the water flows to the Mediterranean sea. We implement the description of the key processes in the land-sea transfer of nutrients, focusing on

nitrogen and phosphate ($NO_3$ and $NH_4$). Model applications should provide a basis for assessment of how nutrient release can vary depending on land management practices and wastewater treatment regulations.

## 2   Model description

### 2.1   Presentation of the LPJmL-Med model

The Lund Potsdam Jena managed Land (LPJmL) is a dynamic global vegetation, hydrology, and crop growth model. The origi-

nal LPJ model (Sitch et al., 2003; Gerten et al., 2004), focused on the modelling of the geographical distribution of nine natural plant functional types (PFTs) with the associated biogeochemical processes (*i.e* carbon cycling and water balance). In order to better assess the role of agriculture within the global climate-vegetation system, Bondeau et al. (2007) introduced the concept of crop functional types (CFTs) to the LPJ model. This model has since then been referred to as LPJmL (Lund–Potsdam–Jena





with managed Land), and simulates the transient changes of terrestrial carbon and water cycles due to land use and agricul-
tural activities under a changing climate. Schaphoff et al. (2018) documented the core LPJmL structure, including equations
and parameters from Sitch et al. (2003) and Bondeau et al. (2007) and all more recent code developments. Lastly, Von Bloh
et al. (2018) extended the LPJmL model to cover the terrestrial nitrogen cycle by explicitly adding processes of soil nitrogen
dynamics, plant uptake, photosynthesis, transpiration, and maintenance respiration to variable nitrogen concentrations in plant
organs, and agricultural nitrogen management.

The LPJmL model has been widely applied to research questions on the terrestrial carbon cycle, hydrology, and agricultural
production (Lutz et al., 2019; Schaphoff et al., 2018, and references therein). It has been tested in multimodel simulations of
productivity and water fluxes (Schewe et al., 2014; Elliott et al., 2015; Müller et al., 2017; Schauberger et al., 2017). One
important characteristic of the LPJmL model is to be dynamic, as the properties of crops can change temporally and are locally
dependent on external forcing. For example, agricultural practices such as sowing date depend on climatic conditions (which
impacts the cultivar choice), and crop management as irrigation practices and residue management can be modified (Bondeau
et al., 2007).

LPJmL basic version uses 13 CFTs (11 annual arable crops and two managed grass types), with specific parameterizations
(Bondeau et al., 2007). Additional CFTs can be added for specific applications (Lapola et al., 2009; Jans et al., 2020). For
Mediterranean studies, Fader et al. (2015) recently included 10 additional crop types in LPJmL, mainly perennial crops, which
are important in the Mediterranean and are often major consumers of irrigation water. With this LPJmL-Med version, the model
can estimate the impacts of different demographic projections and irrigation management systems on production and irrigation
water consumption under climate change for the entire Mediterranean basin (Fader et al., 2016).

So far, LPJmL has not explicitly accounted for the release of nutrients ($NO_3$ and $PO_4$) to the sea from agricultural activities
and urban waste water (via runoff from rivers). In the present study the representation of the nutrient transfer from land to sea is
introduced into LPJmL-Med by considering the following processes: adsorption, remineralization, nitrification, denitrification,
and phytoplankton dynamics (Fig. 1). A compilation of a new input data set of fertilizer, manure and wastewater nutrients
content, at an annual time step for the period [1961-2005] has been established at high spatial resolution grid (1/12°) and over
watersheds of rivers flowing into the Mediterranean Sea (Fig. 2). All implemented processes and the corresponding references
are described in detail in the following sections.

## 115  2.2  General equations

We describe here all new model features in full detail, *i.e* the representation of the nutrient transfer from land to sea, by
considering the following processes: adsorption, remineralization, nitrification, denitrification, phytoplankton dynamics, and
by the implementation of new external input of nutrients. Therefore, we start with general equations describing the nutrient
dynamic in soil and in water (*cf.* below). All parameters and variables used in the model are summarized in Tables A1, A2 and
A3 in the Appendix.

State variables used to describe C, N and P cycles in the LPJmL-Med model are the following:





- Residues content in the soil layer $l$ ($\text{RES}_{C,l}$, $\text{RES}_{N,l}$ and $\text{RES}_{P,l}$ in g).

- Decomposers content in the soil layer $l$ ($\text{DEC}_{C,l}$, in g ), only represented through a carbon content, their N and P contents being deduced from fixed C:N and C:P ratios.

- Nitrate, ammonium and phosphate content in layer $l$ of the soil ($\text{NO3}_l$, $\text{NH4}_l$, $\text{PO4}_l$ in g).

- C biomass of water decomposers ($\text{DEC}_{C,water}$ in $\text{gC·m}^{-3}$).

- Detrital organic matter concentration in water ($\text{RES}_{C,water}$ in $\text{gC·m}^{-3}$).

- Carbon phytoplankton concentration in water ($\text{PHYTO}_C$ in $\text{gC·m}^{-3}$).

- Nitrate, ammonium and phosphate concentrations in water ($\text{NO3}_{water}$, $\text{NH4}_{water}$, $\text{PO4}_{water}$ in $\text{g·m}^{-3}$).

**Soil residues**

The soil residue pool consists of the organic matter coming either from the dead biomass entering the litter pool or from the applied manure. Its mineralization will produce inorganic nutrients. The residue pool is updated for each soil layer, at different time steps depending on the source of the organic matter. Manure is applied in agricultural fields twice a year. For natural PFTs, the litter pool is incremented monthly with the fine root turnover, daily (for evergreen PFTs) or at leaf fall (for deciduous

PFTs) with the leaf turnover, yearly with the wood biomass of killed trees. For CFTs, the litter pool is incremented at harvest with fractions of the different organs depending on the farming practices. Residues in the soil layers below the surface layer are fed by roots only. Part of the residues is consumed by soil decomposers.

The dynamics of carbon, nitrogen and phosphorus ($X$ either stands for N or P) contents of soil residues is described by the following equations for the above soil layer ($l = 0$):

$$\frac{\partial \text{RES}_{C,0}}{\partial t} \quad = \quad \sum_i \text{FRAC}_{res,i} \cdot \text{VEG}_{C,i} - \text{G}_{dec,0} \tag{1}$$

$$\frac{\partial \text{RES}_{X,0}}{\partial t} \quad = \quad \text{MAN}_X + \sum_i \text{FRAC}_{res,i} \cdot \frac{\text{VEG}_{C,i}}{\text{RATIO}_{C:X,i}} - \text{G}_{dec,0} \cdot \frac{\text{RES}_{X,0}}{\text{RES}_{C,0}} \tag{2}$$

and for the deeper soil layers, in contrast to residues added to the first soil layer, those added in underneath soil layers

($1 \leq l \leq 4$) only come from roots :

$$\frac{\partial \text{RES}_{C,l}}{\partial t} \quad = \quad \text{FRAC}_{res,root,l} \cdot \text{ROOT}_{C,l} - \text{G}_{dec,l} \tag{3}$$





$$\frac{\partial \text{RES}_{X,l}}{\partial t} \quad = \quad \text{FRAC}_{res,root,l} \cdot \frac{\text{ROOT}_{C,l}}{\text{RATIO}_{C:X,root}} - \text{G}_{dec,l} \cdot \frac{\text{RES}_{X,l}}{\text{RES}_{C,l}} \tag{4}$$

where $\text{VEG}_{C,i}$ is the C biomass of the plant part $i$ ($i$ = root, sapwood, leaf, storage organ) in the top soil layer and $\text{ROOT}_{C,l}$, the C biomass of roots in layer $l$ ($1 \leq l \leq 4$). See section 2.4.1 for more details. $\text{FRAC}_{res,i}$ is the daily fraction of plant part $i$ entering the residue compartment of the above soil layer due to senescence, mortality, or harvest (section 2.4.1). $\text{FRAC}_{res,root,l}$ ($l > 0$) is the daily fraction of root entering the residue compartment of soil layer $l$ due to senescence, mortality, or harvest. $\text{RATIO}_{C:X,i}$ is the C:X ratio for the plant part $i$. $\text{MAN}_X$ is the content in element $X$ of the daily applied manure (section

2.3.2). $\text{G}_{dec,l}$ is the daily amount of C residues consumed by soil decomposers in layer $l$ (section 2.4.1).

**Soil decomposers dynamics**

Changes in the C biomass of soil decomposers in each soil layer $l$ ($0 \leq l \leq 4$) is assessed as follows:

$$\frac{\partial \text{DEC}_{C,l}}{\partial t} \quad = \quad \text{G}_{dec,l} - m_{dec} \cdot \text{DEC}_C \tag{5}$$

where $m_{dec}$ is the daily mortality rate of decomposers (*cf.* Table. A1).

**Water residues dynamics**

Detrital organic matter consumed by decomposers are substracted from the residue pool in the water:

$$\frac{\partial \text{RES}_{C,water}}{\partial t} \quad = \quad \text{m}_{phyto} \cdot \text{PHYTO}_C - sed \cdot \text{RES}_{C,water} - \text{G}_{dec,water} \tag{6}$$

where $\text{PHYTO}_{C,water}$ is the Phytoplankton C biomass (*cf.* equation 11), $\text{m}_{phyto}$ is the daily mortality of phytoplankton and water decomposers, including grazing, $sed$ is the fraction of residues trapped daily into sediments, $\text{G}_{dec,water}$ is the daily amount of C residues consumed by water decomposers (section 2.4.2).

**Water decomposers dynamics**

Changes in the C biomass of water decomposers ($\text{DEC}_{C,water}$) is assessed as follows:

$$\frac{\partial \text{DEC}_{C,water}}{\partial t} \quad = \quad \text{G}_{dec,water} - \text{m}_{phyto} \cdot \text{DEC}_{C,water} \tag{7}$$





**Nutrient dynamics in soil (daily time step):**

The nutrients (*i.e.* NO$_3$, NH$_4$ and PO$_4$) dynamics in soil can be represented by the following equations:

$$\frac{\partial \text{NO3}_l}{\partial t} = \text{FERT}_{\text{NO3},l} - \text{UPT}_{\text{NO3},l} + \text{NITR}_{\text{NH4},l} - \text{DEN}_{\text{NO3},l} + \text{LEACH}_{\text{NO3},l} \tag{8}$$


$$\frac{\partial \text{NH4}_l}{\partial t} = \text{FERT}_{\text{NH4},l} + \text{REMIN}_{\text{NH4},l} \cdot \text{NH4}_l - \text{UPT}_{\text{NH4},l} - \text{NITR}_{\text{NH4},l} + \text{LEACH}_{\text{NH4},l} \tag{9}$$

$$\frac{\partial \text{PO4}_l}{\partial t} = \text{FERT}_{\text{PO4},l} + \text{REMIN}_{\text{PO4},l} \cdot \text{PO4}_l - \text{UPT}_{\text{PO4},l} - \text{ADS}_{\text{PO4},l} + \text{LEACH}_{\text{PO4},l} \tag{10}$$

Where $\text{FERT}_{\text{NO3},l}$, $\text{FERT}_{\text{NH4},l}$, $\text{FERT}_{\text{PO4},l}$ are the input of NO$_3$, NH$_4$, PO$_4$ synthetic fertilizers in crop fields of the surface layer (*cf.* Section 2.3.1). $\text{REMIN}_{X,l}$, ($X$ = NH$_4$, PO$_4$): remineralization of NH$_4$, and PO$_4$, following the decomposition of organic N and P (from manure and litter residues, sections 2.3.2 and 2.4.1). $\text{UPT}_{\text{NO3},l}$, $\text{UPT}_{\text{NH4},l}$, and $\text{UPT}_{\text{PO4},l}$: uptake of NO$_3$, NH$_4$, PO$_4$ by the plants (Sect 2.4.1). $\text{NITR}_{\text{NH4},l}$: nitrification of NH$_4$ into NO$_3$ in soil (Sect 2.4.4). $\text{DEN}_{\text{NO3},l}$: NO$_3$ denitrification in soil (Sect 2.4.5). $\text{ADS}_{\text{PO4},l}$: Adsorption of PO$_4$ on soil particles lost through sedimentation (Sect 2.4.3). 185    $\text{LEACH}_{\text{NO3},l}$, $\text{LEACH}_{\text{NH4},l}$, and $\text{LEACH}_{\text{PO4},l}$: leaching, *i.e.* transfer of nutrient in the different soil layers = inflow - outflow of nutrient through water percolation and surface/subsurface runoff (Sect 3.2).

**Phytoplankton dynamics**

Phytoplankton C biomass (PHYTO$_C$ in gC) is calculated in equation 11 as:

$$\frac{\partial \text{PHYTO}_C}{\partial t} = -m_{phyto} \cdot \text{PHYTO}_C + \text{G}_{phyto} \cdot \text{PHYTO}_C \tag{11}$$

G$_{phyto}$ is the daily growth rate of Phytoplankton (*cf.* section 2.4.2) and m$_{phyto}$ is the daily mortality of phytoplankton and water decomposers, including grazing.

**Nutrient dynamics in water (daily time step):**

The nutrients (*i.e.* NO$_3$, NH$_4$ and PO$_4$) dynamics in water can be represented by the following equations:

$$\frac{\partial \text{NO3}_{water}}{\partial t} = \text{SEW}_{\text{NO3},water} - \text{UPT}_{\text{NO3},water} + \text{NITR}_{\text{NH4},water} - \text{DEN}_{\text{NO3},water} + \text{TRANS}_{\text{NO3},water} \tag{12}$$






$$\frac{\partial \text{NH4}_{water}}{\partial t} = \text{SEW}_{\text{NH4},water} + \text{REMIN}_{\text{NH4},water} - \text{UPT}_{\text{NH4},water} - \text{NITR}_{\text{NH4},water} + \text{TRANS}_{\text{NO3},water} \tag{13}$$

$$\frac{\partial \text{PO4}_{water}}{\partial t} = \text{SEW}_{\text{PO4},water} + \text{REMIN}_{\text{PO4},water} - \text{UPT}_{\text{PO4},water} - \text{ADS}_{\text{PO4},water} + \text{TRANS}_{\text{NO3},water} \tag{14}$$

Where: $\text{SEW}_{\text{NO3},water}$, $\text{SEW}_{\text{NH4},water}$, $\text{SEW}_{\text{PO4},water}$: input of $NO_3$, $NH_4$, $PO_4$ in rivers and reservoirs from sewage (Sect 2.3.3). $\text{REMIN}_{\text{NH4},water}$, and $\text{REMIN}_{\text{PO4},water}$: remineralization of $NH_4$, $PO_4$, following the decomposition of organic N and P (from phytoplankton residues, Sect 2.4.2). $\text{UPT}_{\text{NO3},water}$, $\text{UPT}_{\text{NH4},water}$, and $\text{UPT}_{\text{PO4},water}$: uptake of $NO_3$, $NH_4$, $PO_4$ by phytoplankton (Sect 2.4.2). $\text{ADS}_{\text{PO4},water}$: Adsorption of $PO_4$ on suspended particles in water (Sect 2.4.3). $\text{NITR}_{\text{NH4},water}$: nitrification of $NH_4$ into $NO_3$ in water (Sect 2.4.4). $\text{DEN}_{\text{NO3},water}$: $NO_3$ denitrification in water (Sect 2.4.5). $\text{TRANS}_{\text{NO3},water}$,

$\text{TRANS}_{\text{NH4},water}$, and $\text{TRANS}_{\text{PO4},water}$: transport, *i.e.* transport of nutrients along water flow performed through river routine (*cf.* Sect 3).

Note: (i) Nitrogen-fixation is implicitly represented by considering the N uptake from N-fixers (the legumes soybeans and pulses) comes from N-fixers only, resulting in $\text{UPT}_{\text{NO3}} = \text{UPT}_{\text{NH4}} = 0$ for those CFTs. Therefore, there is no specific Nitrogen-fixation element within the global equations (ii) Although $NO_x$ deposition has clearly increased in the last decades, its values

in the Mediterranean remain low (apart in the Po region) when compared to the load from agriculture. It is not included in the equations.

### 2.3    Implementation of external inputs of nutrients

In this section, we will focus on processes leading to external inputs of nutrients to soils and rivers, either associated with agricultural practices (*i.e.*, application of manure and inorganic fertilizers to soils) or with household wastewater release.

#### 2.3.1    Fertilizers application

N and P fertilizers are applied to the first soil layer and depend on the country, the crop type and the cultivated area. N and P fertilizers are applied twice or thrice each year depending on the crop fertilization calendar (Table 1). The IFADATA database (IFA, 2016) provides N and P fertilizer data, we consider that half of the inorganic N inputs from fertilizers enters the nitrate pool in the first soil layer, and the remaining half feeds the ammonium pool. On fertilizer days, we estimate the amount of ,

NH4, and PO4fertilizers applied to each stand on the first soil layer (*l*=0), though equation 15 as in Potter et al. (2010):



$$\text{FERT}_{NO3,l=0} = \begin{cases} \text{FRAC}_{fert} \cdot 1/2 \cdot \text{FERT}_{N,\text{IFA}} \cdot \frac{\text{AREA}}{\text{AREATOT}} & \text{for days with fertilization} \\ 0 & \text{for other days} \end{cases}$$

$$\text{FERT}_{NH4,l=0} = \begin{cases} \text{FRAC}_{fert} \cdot 1/2 \cdot \text{FERT}_{N,\text{IFA}} \cdot \frac{\text{AREA}}{\text{AREATOT}} & \text{for days with fertilization} \\ 0 & \text{for other days} \end{cases}$$

$$\text{FERT}_{PO4,l=0} = \begin{cases} \text{FRAC}_{fert} \cdot \text{FERT}_{P,\text{IFA}} \cdot \frac{\text{AREA}}{\text{AREATOT}} & \text{for days with fertilization} \\ 0 & \text{for other days} \end{cases}$$

(15)

$\text{FERT}_{N,\text{IFA}}$ and $\text{FERT}_{P,\text{IFA}}$ are the annual inputs of N and P fertilizers obtained from the IFADATA database IFA (2016) for each country. AREA is the area of the stand, while AREATOT is the total cultivated area in the country, both obtained from the land use input data of the LPJmL-Med model (*cf.* Fig. 3). $\text{FRAC}_{fert}$ corresponds to the fraction of fertilizers applied each
fertilizer day (1/2, 1/3 or 0 depending of crop type, 1).

### 2.3.2 Manure application

As for inorganic fertilizers, manure is applied to the first soil layer and depends on the country, the crop type and the cultivated area. As it is composed of organic matter, manure is added to the litter (see Fig. 1). To estimate the amount of N manure introduced in each stand ($\text{MAN}_N$), on days when this occurs, we distinguish between pastures and other crops (*cf.* equation
230 16):

$$\text{MAN}_N = \begin{cases} \text{FRAC}_{fert} \cdot \text{MAN}_{N,lc} \cdot \frac{\text{AREA}}{\text{AREATOT}_{lc}} & \text{for days with manure application} \\ 0 & \text{for other days} \end{cases}$$

(16)

$lc$ in equation 16 refers to the land cover which here can either be pasture or other cultivated areas (i.e. no pasture). $\text{MAN}_{N,\text{pasture}}$ and $\text{MAN}_{N,\text{nopasture}}$ are the annual amounts of N manure left on pastures and applied to crops, respectively, obtained from FAOSTAT (FAO 2016a, 2016b) for each country. $\text{AREATOT}_{\text{pasture}}$ and $\text{AREATOT}_{\text{nonpasture}}$ correspond to the
total area of pastures and other cultivated areas, respectively, in the country, obtained from the land use input data of the LPJmL-Med model (Fig. 4). In contrast to inorganic fertilizers, we consider that manure is applied twice each year for all crop types except pastures where it is supposed to be spread) uniformly over the grazing period 6 months long in the northern hemisphere, all year long in the southern hemisphere, as described in the crop fertilization calendar (Table 1). As data for P manure application are not available from the FAOSTAT database, we estimate it using a constant country-specific ($\text{P}_{MAN} : \text{N}_{MAN}$)
ratio provided by Potter et al., (2011a, 2011b) (*cf.* equation 17):

$$\text{MAN}_P = \text{MAN}_N \cdot (\text{P}_{MAN} : \text{N}_{MAN})$$

(17)





where $P_{MAN}$ and $N_{MAN}$ in equation 17 are the total N and P in manure produced in 1994-2001, obtained from Potter et al., (2011a, 2011b).

### 2.3.3 Wastewater release

Wastewater is released into rivers and reservoirs (Fig. 1, Fig. 5). We distinguish the emissions by urban and rural populations, as they can have an uneven access to wastewater treatment plants (Van Drecht et al., 2003, 2009). The daily point source of N and P from sewer systems (SEW$_X$, $X$ =N, P) is estimated following equations 18 from Van Drecht et al. (2003, 2009):

$$\text{SEW}_X = \text{WW}_{X,urban} \cdot \text{POP}_{urban} + \text{WW}_{X,rural} \cdot \text{POP}_{rural} \tag{18}$$

WW$_{X,urban}$ and WW$_{X,rural}$ ($X$ =N, P) correspond to the annual per capita X release from wastewater in urban and rural

areas. Urban and rural population size in each grid cell (POP$_{urban}$ and POP$_{rural}$, respectively) are provided each ten years at 5' spatial resolution by the HYDE3.2 database (Klein Goldewijk et al., 2011). Half of the N released into the rivers and reservoirs is considered to be nitrate, and half ammonium:

$$\text{SEW}_{\text{NO3},water} = \frac{\text{SEW}_{N,water}}{2} \qquad \text{and} \qquad \text{SEW}_{\text{NH4},water} = \frac{\text{SEW}_{N,water}}{2} \tag{19}$$

The N content of wastewater is determined only by human emissions ($E_{N,hum}$), while P release is determined by both human

emissions ($E_{P,hum}$) and emissions from laundry and dishwasher detergent ($E_{P,Ldet}$ and $E_{P,Ddet}$, respectively) as shown in equation 20 (Van Drecht et al., 2003, 2009):

$$
\begin{aligned}
\text{WW}_{N,urban} &= \text{E}_{N,hum} \cdot \text{FRAC}_{sew,urban} \cdot \frac{\sum\limits_{i=0}^{3} \text{FRAC}_{sew,i} \cdot (1 - \text{REM}_{N,i})}{\sum\limits_{i=0}^{3} \text{FRAC}_{sew,i}} \\[2em]
\text{WW}_{N,rural} &= \text{E}_{N,hum} \cdot \text{FRAC}_{sew,rural} \cdot \frac{\sum\limits_{i=0}^{3} \text{FRAC}_{sew,i} \cdot (1 - \text{REM}_{N,i})}{\sum\limits_{i=0}^{3} \text{FRAC}_{sew,i}} \\[2em]
\text{WW}_{P,urban} &= \left( \text{E}_{P,hum} + \text{E}_{P,Ldet} + \frac{E_{P,Ddet}}{\text{FRAC}_{sew,urban}} \right) \cdot \text{FRAC}_{sew,urban} \cdot (1 - \text{REM}_{P,1}) \\[1em]
\text{WW}_{P,rural} &= \left( \text{E}_{P,hum} + \text{E}_{P,Ldet} + \frac{E_{P,Ddet}}{\text{FRAC}_{sew,rural}} \right) \cdot \text{FRAC}_{sew,rural} \cdot (1 - \text{REM}_{P,1})
\end{aligned}
\tag{20}
$$

FRAC$_{sew,urban}$ and FRAC$_{sew,rural}$ in equations 20 correspond to the fraction of the urban and rural population that is connected to public sewerage system. Four categories of treatment are considered: no treatment ($i = 0$), primary/mechanistic treatment ($i = 1$), secondary/biological treatment ($i = 2$), and tertiary/advanced treatment ($i = 3$). The fraction of the total population connected to public sewerage system with treatment $i$ is determined by FRAC$_{sew,i}$ . REM$_{N,i}$ and REM$_{P,i}$ correspond





to the fractions of N and P removed from wastewater with treatment $i$ (*cf.* Table A1). When no data is available concerning the access of population to the different types of sewerage systems, we consider that population can only access primary treatment.

The per capita human emissions of N ($E_{N,hum}$) and P ($E_{P,hum}$) are estimated from the national per capita gross domestic product based on purchasing power parity ($GDP_{PPP}$) as shown in equations 21 (Van Drecht et al., 2003, 2009):

$$
\begin{aligned}
E_{N,hum} &= 365 \cdot \left[ 4 + 14\left(\frac{GDP_{PPP}}{33000}\right)^{0.3} \right] \\
E_{P,hum} &= \frac{E_{N,hum}}{RATIO_{N:P,ww}}
\end{aligned}
\tag{21}
$$

where $RATIO_{N:P,ww}$ is the mean N:P ratio of municipal wastewater.

The per capita emission of P from laundry and dishwasher detergents ($E_{P,Ldet}$ and $E_{P,Ddet}$, respectively in equations 22) are estimated from socioeconomic indices (Van Drecht et al., 2009):

$$
\begin{aligned}
E_{P,Ldet} &= CONT_{P,Ldet} \cdot (1 - FRAC_{Pfree,Ldet}) \cdot \left[ 10 - 10\left(\frac{GDP_{mer}}{20000} - 1\right)^2 \right] \\
E_{P,Ddet} &= CONT_{P,Ddet} \cdot \frac{CONS_{hh,Ddet}}{PPHH}\left( 0.25 + 0.07\frac{GDP_{mer}}{10000} \right)
\end{aligned}
\tag{22}
$$

$CONT_{P,Ldet}$ and $CONT_{P,Ddet}$ are the P content of laundry and dishwasher detergents. $FRAC_{Pfree,Ldet}$ is the fraction of P-free laundry detergents. The fraction of P-free dishwasher detergents is assumed to be negligible (Van Drecht et al., 2009). $GDP_{mer}$ corresponds to the national per capita gross domestic product based on market exchange rate. $CONS_{hh,Ddet}$ corresponds to the mean consumption of dishwasher detergents for each household that owns a dishwasher. PPHH is the average number of persons by household.

The fraction of P-free laundry detergent is estimated in equation 23 as in Van Drecht et al. (2009), and depends on the period concerned:

- In year $yr$ between 1960 and 1999:

$$
FRAC_{Pfree,Ldet}(yr) = FRAC_{Pfree,Ldet}(2000) \cdot \left(1 + \frac{1}{e^{a(2000-b)}}\right) \cdot \left(1 - \frac{1}{e^{a(yr-b)}}\right)
\tag{23}
$$

$FRAC_{Pfree,Ldet}(2000)$ in equation 23 corresponds to the fraction of P-free laundry detergents in year 2000 (calculated from equation 24). a and b are fixed parameters (a = 0.2 and b = 1990), estimated by Van Drecht et al. (2009).

- In 2000 and afterwards:

$$
FRAC_{Pfree,Ldet}(yr) = \min\left(\frac{GDP_{mer}}{33000}, 1\right)
\tag{24}
$$





## 2.4 Transformation and retention processes

In this section, we will focus on processes leading either to a change in the form of nutrients (*i.e.* primary production, residue decomposition and nitrification) or to a loss of nutrients from soils and rivers (*i.e.* sedimentation, adsorption and denitrification).

### 2.4.1 Primary production and residue decomposition in terrestrial systems

Both natural and cultivated plants consume nitrate, ammonium and phosphate in soils. Unlike more complex modelling of the nutrient uptake (*e.g.* Von Bloh et al. 2018 for N), we consider here a simple formulation, similar for N and P, in order to estimate their daily uptake by plants. As the growth of all plant parts (*i.e.* leaves, roots, sapwood and storage parts) is already simulated in the LPJmL-Med model in terms of C, we use the specific stoichiometry of the different plant parts to infer the corresponding daily uptake of nitrate, ammonium and phosphate ($UPT_{NO3}$, $UPT_{NH4}$ and $UPT_{PO4}$, respectively, *cf* equations 25). In the LPJmL-Med model, the soil is divided in 5 layers (numbered from 0 to 4) in order to represent the vertical heterogeneity in water and carbon content of soils. Nutrient uptake by plants in the soil layer $l$ is then assumed to depend on the fraction of total roots of this soil layer ($FRAC_{root,l}$):

$$
\begin{aligned}
UPT_{NO3,l} &= \frac{NO3_l}{NO3_l + NH4_l} \cdot \left( \frac{UPT_{C,root} \cdot FRAC_{root,l}}{RATIO_{C:N,root}} + \frac{UPT_{C,sap}}{RATIO_{C:N,sap}} + \frac{UPT_{C,leaf}}{RATIO_{C:N,leaf}} + \frac{UPT_{C,stor}}{RATIO_{C:N,stor}} \right) \\
UPT_{NH4,l} &= \frac{NH4_l}{NO3_l + NH4_l} \cdot \left( \frac{UPT_{C,root} \cdot FRAC_{root,l}}{RATIO_{C:N,root}} + \frac{UPT_{C,sap}}{RATIO_{C:N,sap}} + \frac{UPT_{C,leaf}}{RATIO_{C:N,leaf}} + \frac{UPT_{C,stor}}{RATIO_{C:N,stor}} \right) \\
UPT_{PO4,l} &= \frac{UPT_{C,root} \cdot FRAC_{root,l}}{RATIO_{C:P,root}} + \frac{UPT_{C,sap}}{RATIO_{C:P,sap}} + \frac{UPT_{C,leaf}}{RATIO_{C:P,leaf}} + \frac{UPT_{C,stor}}{RATIO_{C:P,stor}}
\end{aligned}
\tag{25}
$$

$UPT_{C,i}$ is the daily C uptake for the growth of plant part $i$ ($i$ = root, sapwood, leaf, storage) derived from the LPJmL-Med model. $RATIO_{C:N,i}$ and $RATIO_{C:P,i}$ correspond to the C:N and C:P ratios of plant part $i$ (Mooshammer et al., 2014). Note that for crops associated with N-fixers (*i.e.* soybean and pulses), all the N uptake is assumed to come from symbiotic N-fixers, thus $UPT_{NH4} = UPT_{NO3} = 0$.

Part of the plant biomass is added to the soil residue pool at senescence. This can also be the case at harvest depending on the crop residue management. Residues added to the first soil layer ($l = 0$) correspond to the roots in that layer, as well as to dead organic material from the above-ground parts of the plants (equations 26):





$$\sum_i \text{FRAC}_{res,i} \cdot \text{VEG}_{C,i} = \text{FRAC}_{res,root} \cdot \text{ROOT}_{C,0} + \text{FRAC}_{res,sap} \cdot \text{SAP}_C + \text{FRAC}_{res,leaf} \cdot \text{LEAF}_C + \text{FRAC}_{res,stor} \cdot \text{STOR}_C$$

$$\sum_i \text{FRAC}_{res,i} \cdot \text{VEG}_{N,i} = \text{FRAC}_{res,root} \cdot \frac{\text{ROOT}_{C,0}}{\text{RATIO}_{C:N,root}} + \text{FRAC}_{res,sap} \cdot \frac{\text{SAP}_C}{\text{RATIO}_{C:N,sap}} + \text{FRAC}_{res,leaf} \cdot \frac{\text{LEAF}_C}{\text{RATIO}_{C:N,leaf}}$$

$$+ \text{FRAC}_{res,stor} \cdot \frac{\text{STOR}_C}{\text{RATIO}_{C:N,stor}}$$

$$\sum_i \text{FRAC}_{res,i} \cdot \text{VEG}_{P,i} = \text{FRAC}_{res,root} \cdot \frac{\text{ROOT}_{C,0}}{\text{RATIO}_{C:P,root}} + \text{FRAC}_{res,sap} \cdot \frac{\text{SAP}_C}{\text{RATIO}_{C:P,sap}} + \text{FRAC}_{res,leaf} \cdot \frac{\text{LEAF}_C}{\text{RATIO}_{C:P,leaf}}$$

$$+ \text{FRAC}_{res,stor} \cdot \frac{\text{STOR}_C}{\text{RATIO}_{C:P,stor}}$$

$$(26)$$

$\text{VEG}_{X,i}$ is the $X$ ($X = C, N, P$) biomass of the plant part $i$ ($i$ = root, sapwood, leaf, storage organ). $\text{FRAC}_{res,i}$ is the daily
fraction of plant part $i$ entering the residue compartment of the above soil layer due to senescence, mortality, or harvest. $\text{SAP}_C$,
$\text{LEAF}_C$ and $\text{STOR}_C$ are the total amount of C in roots, sapwood, leaves and storage parts of the plant, respectively. $\text{ROOT}_C$ is
the turnover fraction of C in root, $\text{FRAC}_{res,i}$ corresponds to the fraction of plant part $i$ entering the residue compartment due
to senescence or harvest. In the case of non-permanent crops, $\text{FRAC}_{res,i}$ depends on residue management practices.

Part of the residues is consumed by soil decomposers. We assume that residue consumption in soils follows a donor-
controlled functional response (Zheng et al., 1997), and thus is independent of decomposer biomass. As soil decomposers
have a different stoichiometry from residues, their growth will be limited by the scarcest nutrient (either N, P or C). In order
to maintain homeostasy (e.g. Daufresne and Loreau, 2001), soil decomposers can immobilize part of the inorganic nutrients
available in the soil (equation 27):

$$G_{dec,l} = \min\Big(\text{VMAX}_{res,l}\ \text{S}(24,14)\ \text{RES}_{C,l}\ ,$$

$$\textit{IMMO}_N \cdot (\text{NO3}_l + \text{NH4}_l) \cdot \frac{\text{RES}_{C,l}}{\dfrac{\text{RES}_{C,l}}{\text{RATIO}_{C:N,dec}} - \text{RES}_{N,l}}\ ,$$

$$\text{IMMO}_P \cdot \text{PO4}_l \cdot \frac{\text{RES}_{C,l}}{\dfrac{\text{RES}_{C,l}}{\text{RATIO}_{C:P,dec}} - \text{RES}_{P,l}}\Big)$$

$$(27)$$

$G_{dec,l}$ is the daily amount of C residues consumed by soil decomposers in layer $l$ ($1 \leq l \leq 4$). $\text{VMAX}_{res,l}$ corresponds
to the maximum fraction of C residues consumed daily by soil decomposers (Schjonning et al., 2004). $\text{S}(T_{opt}, \gamma)$ defines a
bell shaped temperature dependence of soil decomposition (see Billen et al. 1994), as described in equation 28. $\text{IMMO}_X$
is the maximum fraction of inorganic nutrient $X$ ($X = N, P$) immobilized daily by soil decomposers to meet their nutrient
requirements. $\text{RATIO}_{C:N,dec}$ and $\text{RATIO}_{C:P,dec}$ are the C:N and C:P ratios of soil decomposers, respectively.





The bell shaped temperature-dependence of biological processes is determined as follows:

$$S(T_{opt}, \gamma) = exp\Big( - \frac{(T_{water} - T_{opt})^2}{\gamma^2} \Big) \tag{28}$$

where $T_{opt}$ is the optimal temperature for the process considered, $\gamma$ is the sigmoid range of T, and $T_{env}$ is the environment
temperature (*i.e.* soil or water temperature depending on the process considered).

Finally, the remineralization $REMIN_{X,l}$, ($X = NH_4$ and $PO_4$) following the decomposition of organic N and P is determined
as follows:

$$REMIN_{X,l} \quad = \quad m_{dec} \cdot \frac{DEC_{C,l}}{RATIO_{C:X,dec}} \tag{29}$$

where $m_{dec}$ is the daily mortality of decomposers.

### 2.4.2   Primary production, sedimentation and residue decomposition in rivers and reservoirs

Phytoplankton consumes nitrate, ammonium and phosphates in rivers and reservoirs. We consider that phytoplankton consumes
nitrate and ammonium in the same proportion as they are available in water. We assume that Liebig's law of the minimum (Von
Liebig, 1942) governs the growth of phytoplankton, *i.e.* that its daily growth rate ($G_{phyto}$) is limited by either N or P as shown
in equation 30:

$$G_{phyto} = \mu_{phyto} \cdot S(24, 14) \cdot \min \Big( \frac{NO3_{water} + NH4_{water}}{NO3_{water} + NH4_{water} + K_{N,phyto} V_{water}} , \frac{PO4_{water}}{PO4_{water} + K_{P,phyto} V_{water}} \Big) \tag{30}$$

$\mu_{phyto}$ is the maximum daily growth of phytoplankton (in $d^{-1}$). $K_{N,phyto}$ in $gN/m^3$ and $K_{P,phyto}$ in $gP/m^3$ correspond to the
half-saturation constants for phytoplankton growth regarding N and P, respectively. $S(Topt, \gamma)$ is already described in equation
28. $V_{water}$ is the volume of water in the river or reservoir considered (in $m^3$). $NO3_{water}$, $NH4_{water}$ and $PO4_{water}$ are the
concentrations of nitrate, ammonium and phosphate, respectively, in water.

Part of the remaining residues are consumed by water decomposers. We assume that organic matter consumption in water
follows a donor-controlled functional response, and thus is independent of decomposer biomass (see equation 31):

$$G_{dec,water} = VMAX_{res,water} \cdot S(24, 14) \cdot RES_{water} \tag{31}$$

$G_{dec,water}$ in equation 31 is the daily consumption of carbon residues by water decomposers. $VMAX_{res,water}$ corresponds to
the maximum fraction of residues consumed daily by water decomposers.

The uptake in inorganic nutrient content of water by phytoplankton is calculated by the following equations 32:





$$
\begin{aligned}
\text{UPT}_{\text{NO3},water} &= \text{G}_{phyto} \cdot \frac{\text{PHYTO}_C}{\text{RATIO}_{C:N,phyto}} \cdot \frac{\text{NO3}_{water}}{\text{NO3}_{water} + \text{NH4}_{water}} \\
\text{UPT}_{\text{NH4},water} &= \text{G}_{phyto} \cdot \frac{\text{PHYTO}_C}{\text{RATIO}_{C:N,phyto}} \cdot \frac{\text{NH4}_{water}}{\text{NO3}_{water} + \text{NH4}_{water}} \\
\text{UPT}_{\text{PO4},water} &= \text{G}_{phyto} \cdot \frac{\text{PHYTO}_C}{\text{RATIO}_{C:P,phyto}}
\end{aligned}
$$

365 (32)

where $\text{RATIO}_{C:N,phyto}$ and $\text{RATIO}_{C:P,phyto}$ are the C:N and C:P mean ratios of phytoplankton cells. Note that we consider that water decomposers and phytoplankton have the same characteristics regarding their mortality rate and stoichiometry.

The remineralization in water $\text{REMIN}_{X,water}$, ($X = \text{NH}_4$ and $\text{PO}_4$) following the phytoplankton water decomposers is determined as follows:

$$ \text{REMIN}_{X,water} = m_{phyto} \cdot \frac{\text{DEC}_{C,water}}{\text{RATIO}_{C:X,phyto}} $$ (33)

### 2.4.3 Adsorption

Part of the inorganic nutrients get adsorbed on particles and are lost through sedimentation, both in soils and rivers (e.g Thieu et al., 2009). Adsorption concerns only $PO_4$, due to its chemical properties. As the efficiency of the adsorption process is 375 strongly dependent on the nutrient concerned, it has an important role in determining the relative proportion of nutrients in both terrestrial and aquatic systems. The daily amounts of $PO_4$ adsorbed on soil particules ($ADS_{PO4,l}$), is computed from equation 34:

$$ \text{ADS}_{PO4,l} = a \cdot \text{m}_l \left( \frac{\text{PO4}_l \cdot}{\text{VOL}_{water,l}} \right)^{b/a} $$ (34)

Where: a and b are fixed parameters (*cf.* table A1). $\text{m}_l$ is the mass of soil in layer $l$. $\text{VOL}_{water,l}$ corresponds to the volume 380 of water in the soil.

In rivers, we consider that the amount of $\text{NH}_4$ adsorbed on particles is negligible. The daily amount of $PO_4$ adsorbed in rivers and reservoirs ($ADS_{PO4,water}$) follows a Michaelis-Menten dynamics and depends on the concentration of suspended particles in the water ($C_{susp,water}$), as described in Billen et al. (2007) (see equation 35):

$$ \text{ADS}_{PO4,water} = \text{C}_{susp,water} \cdot \text{VOL}_{water} \cdot \frac{\text{VMAX}_{PO4,ads} \cdot [\text{PO4}]_{water}}{[\text{PO4}]_{water} + K_{water,PO4} \cdot \text{VOL}_{water}} $$ (35)

$\text{VMAX}_{PO4,ads}$ corresponds to the maximal adsorption rate of $PO_4$ on suspended particles in rivers and reservoirs. $\text{Kads}_{water,PO4}$ is the half saturation constant for $PO_4$ adsorption in rivers and reservoirs.





As land use impacts the stability of soils and the amount of organic material released in aquatic systems, it also affects $C_{susp,water}$ the concentration of suspended particles in rivers and reservoirs (Billen et al., 2007) as shown in equation 36:

$$C_{susp,water} = \frac{C_{susp,forest} \cdot \text{SURF}_{forest} + C_{susp,grass} \cdot \text{SURF}_{grass} + C_{susp,crops} \cdot \text{SURF}_{crops} + C_{susp,urban} \cdot \text{SURF}_{urban}}{\text{SURF}_{cell}}$$

(36)

$C_{susp,i}$ is the concentration of suspended solids in the water released from system $i$ *(i = forests, grasslands, crops, urban areas)*, $\text{SURF}_i$ corresponds to the surface of system $i$ in the cell, whose total surface is $\text{SURF}_{cell}$. The amounts of $PO_4$ lost through adsorption in soils and water is then substracted from their respective $PO_4$ pool.

### 2.4.4 Nitrification

Nitrification is an important process in both terrestrial and aquatic nitrogen cycles, corresponding to the transformation of $NH_4$
into $NO_3$. In each soil layer $l(l = 0 - 4)$, the daily amount of $NH_4$ transformed through nitrification ($\text{NITR}_l$ in gN) depends on soil moisture, temperature and pH (Von Bloh et al., 2018; Parton et al., 1996):

$$\text{NITR}_l = \text{NH4}_l \cdot \text{VMAX}_{soil,nitr} \cdot \text{RESP}_{nitr,moist} \cdot \text{S}(18.79, 7.44) \cdot \text{RESP}_{nitr,pH}$$

(37)

$\text{VMAX}_{soil,nitr}$ is the maximum fraction of $NH_4$ nitrified daily. $\text{RESP}_{nitr,moist}$ and $\text{RESP}_{nitr,pH}$ correspond to the response functions of soil nitrification regarding soil moisture and pH, respectively. $\text{S}(18.79, 7.44)$ defines a bell shaped temperature
dependence of soil nitrification, as described in equation 28 (Von Bloh et al., 2018; Parton et al., 1996).

The response function of soil nitrification regarding soil moisture is calculated as in Von Bloh et al. (2018); Parton et al. (1996) (see equation 38):

$$\text{RESP}_{nitr,moist} = \left( \frac{\text{FRAC}_{soil,water} - b}{a - b} \right)^{\frac{d \cdot (b-a)}{(a-c)}} \cdot \left( \frac{\text{FRAC}_{soil,water} - c}{a - c} \right)^{d}$$

(38)

$\text{FRAC}_{soil,water}$ corresponds to the ratio of soil moisture to soil maximum moisture. $a, b, c$ and $d$ (0.60, 1.27, 0.0012, and 2.84
respectively) are fixed parameters for sandy and medium soils (Von Bloh et al., 2018; Parton et al., 1996).

The response function of nitrification regarding soil pH ($\text{pH}_{soil}$) is based on Parton et al. (1996) (see equation 39):

$$\text{RESP}_{nitr,pH} = 0.56 + \frac{\arctan(0.45 \cdot \pi \cdot (pH_{soil} - 5))}{\pi}$$

(39)

In rivers and reservoirs, nitrification is modeled through a Michealis-Menten equation, as in the Riverstrahler model (Billen et al., 1994), equation 40:

$$\text{NITR}_{water} = \text{VMAX}_{water,nitr} \cdot \text{S}(24, 6) \cdot \frac{\text{NH4}_{water}}{\text{NH4}_{water} + K_{water,nitr}}$$

(40)





$\text{NITR}_{water}$ corresponds to the daily amount of $NH_4$ nitrified in water. $\text{VMAX}_{water,nitr}$ is the maximum nitrification rate in water. $K_{water,nitr}$ is the half-saturation constant for water nitrification. $S(24, 6)$ defines a bell shaped temperature dependence of nitrification (Billen et al., 1994), as described in equation 28.

The amount of N nitrified in soils and water is then added to their respective $NO_3$ pools and substracted from the $NH_4$ pools.

## 2.4.5 Denitrification

Denitrification is a key process occuring along the land to sea N transfer, as it allows the release of N into the atmosphere in the form of dinitrogen. It occurs in hypoxic or anoxic areas, both in soils and aquatic systems. As oxygen availability is not explicitly represented in our model, we use proxies to determine the strength of denitrification in soils and water.

In each soil layer $l(l = 0 - 4)$, the daily amount of N lost through denitrification ($\text{DEN}_l$) depends on soil moisture, temperature and organic content (Von Bloh et al., 2018; Parton et al., 1996), equation 41:

$$
\begin{aligned}
\text{VMAX}_{den,l} &= \text{RESP}_{den,moist} \cdot \left[1 - \exp\left(-1.2 \cdot \text{RES}_{C,l} \cdot \text{RESP}_{den,temp}\right)\right] \\
\text{DEN}_l &= \text{NO3}_l \cdot \text{VMAX}_{den,l}
\end{aligned}
\tag{41}
$$

where: $\text{VMAX}_{den,l}$ is the maximum daily denitrification in soil. $\text{RESP}_{den,moist}$ and $\text{RESP}_{den,temp}$ are the response functions of soil denitrification regarding soil moisture and soil temperature, respectively. $RES_{C,l}$ is the mass of carbon in residues of the soil layer $l$.

In soils, we consider that the oxygen content of a soil layer is inversely related to soil moisture (Von Bloh et al., 2018; Parton et al., 1996):

$$
\text{RESP}_{den,moist} = 6.664096 \cdot 10^{-10} \cdot \exp\left(21.12912 \cdot \text{FRAC}_{soil,water}\right)
\tag{42}
$$

The response function of soil denitrification regarding soil temperature ($T_{soil}$) is assumed to be ascending up to a temperature threshold of 45.9°C, and zero beyond (Von Bloh et al., 2018; Parton et al., 1996), as suggested by equation 43:

$$
RESP_{den,temp} = \begin{cases} 0.0326 & \text{for } T_{soil} \leqslant 0°C \\ 0.00351 \cdot (T_{soil})^{1.652} - \left(\frac{T_{soil}}{41.748}\right)^{7.19} & \text{for } 0°C < T_{soil} < 45.9°C \\ 0 & \text{for } T_{soil} \geqslant 45.9°C \end{cases}
\tag{43}
$$

In rivers and reservoirs, we consider that the oxygen content of water is positively correlated to the hydraulic load ($\text{LOAD}_{water}$ in m/d), thus the strength of denitrification varies inversely with the hydraulic load (Wollheim et al., 2008):

$$
\begin{aligned}
\text{VMAX}_{den,water} &= \text{S}(24,14) \cdot \left[1 - \exp\left(-\frac{\text{VIT}_{water,den}}{\text{LOAD}_{water}}\right)\right] \\
\text{DEN}_{\text{NO3},water} &= \text{NO3}_{water} \cdot \text{VMAX}_{den,water}
\end{aligned}
\tag{44}
$$





$\text{VMAX}_{den,water}$ is the maximum daily denitrification in water. $\text{DEN}_{NO3,water}$ corresponds to the daily amount of nitrogen lost through denitrification in rivers and reservoirs. S(24, 14) defines a bell shaped temperature dependence of denitrification (Billen et al., 1994), as described in equation 28. $\text{VIT}_{water,den}$ is the uptake rate for denitrification in rivers and reservoirs. The amount of N denitrified in soils and water is then substracted from their respective $NO_3$ pools.

## 3 Transport processes

### 3.1 Rivers, dams, and lakes

LPJmL-Med divides the soil column into five hydrological active layers of 0.2, 0.3, 0.5, 1, and 1 m thickness (Sibyll Schaphoff et al., 2013). Water holding capacity and hydraulic conductivity are derived for each grid cell using soil texture from the Harmonized World Soil Database (Nachtergaele et al., 2014) and relationships between texture and hydraulic properties from Cosby et al. (1984). Water content in soil layers is altered by infiltrating rainfall, gravity (percolation), and the plant water uptake. The infiltration rate of rain and irrigation water into the soil depends on the current soil water content of the first layer. The surplus water that does not infiltrate is assumed to generate surface run-off. The lateral exchange of water discharge between grid cells through the river is computed via the river routine module implemented in LPJmL-Med model. The transport of water in the river channel is approximated by a cascade of linear reservoirs (Schaphoff et al., 2018; Rost et al., 2008).

The human influence on the hydrological cycle is explicitly represented in LPJmL-Med model by accounting for irrigation, water consumption, water abstraction, as well as an implementation of dams and reservoirs. Reservoirs are filled daily with discharge from upstream locations and with local precipitation. Cells receive water from the reservoirs when the following conditions are met: (i) cells have a lower altitude than the cell containing the reservoir, and (ii) they are situated along the main river downstream or at maximum five cells upstream. Hence, a cell can receive water from multiple reservoirs. Dams built for irrigation are assumed to release their water proportionally to gross irrigation water demand downstream. Dams built for other purposes (*e.g* flood control, hydro-power) are assumed to release a constant water volume throughout the year. The actual release from a reservoir is simulated to depend on its storage capacity relative to its inflow. If an irrigation purpose is defined for the reservoir, part of the outflow is diverted to irrigated lands downstream. Both surface and subsurface run-off are simulated to accumulate to river discharge (for more detail see Schaphoff et al., 2018).

### 3.2 Nutrient leaching and transport

Nutrients are assumed to be fully dissolved in water and move with lateral runoff, surface runoff, and percolation water. The first step to calculate the quantity of nutrients transported with the water from a soil layer, is to compute the concentration of nutrients in the mobile water. This concentration is then multiplied by the volume of runoff or percolation water between soil layers or into the aquifer. The quantity of nutrients leached depends on climate conditions (*e.g* precipitation), soil conditions and the intensity of soil management (*e.g* fertilization, plant cover, soil treatment). Nutrient movement with water fluxes is





simulated as in SWAT (Neitsch et al., 2002, 2005), and in Von Bloh et al. (2018) in LPJmL5.0 version. The concentration of nutrients (Nut) in the mobile water (in kg m$^{-3}$) is calculated from equation 45:

$$\mathrm{NUT}_{mobile,l} = \frac{\mathrm{NUT}_l \cdot \left(1 - \exp\left(\frac{-\mathrm{w}_{mobile,l}}{(1-\theta)\cdot\mathrm{SAT}_l}\right)\right)}{\mathrm{w}_{mobile,l}} \tag{45}$$

$\mathrm{w}_{mobile}$ is the amount of mobile water in the layer (mm), $\theta = 0.4$ is the fraction of porosity from which anions are excluded

(the same as in Von Bloh et al. 2018, 0.5 in Neitsch et al. 2002), and $\mathrm{SAT}_l$ is the saturated water content of the soil layer (mm). The mobile water $\mathrm{w}_{mobile,l}$ in the layer $l$ is the quantity of water lost by surface runoff, lateral flow, and percolation as shown in equation 46:

$$\mathrm{w}_{mobile,l} = \begin{cases} \mathrm{Q}_{surf} + \mathrm{Q}_{lat,l=0} + \mathrm{w}_{perc,l=0} & \text{for } l = 0 \\ \mathrm{Q}_{lat,l} + \mathrm{w}_{perc} & \text{for } l > 0 \end{cases} \tag{46}$$

where $\mathrm{Q}_{surf}$ is the surface runoff (only in the top soil layer; mm), $\mathrm{Q}_{lat,l}$ is the water discharged from the layer by lateral flow

(mm), and $\mathrm{w}_{perc,l}$ is the amount of water percolating to the underlying soil layer on a given day (mm).

Finally, the flux of nitrate that is removed through surface runoff $\mathrm{Fnut}_{surf}$ and lateral flow $\mathrm{FNut}_{lat,l}$ is calculated in equation 47 as:

$$\begin{aligned} \mathrm{Fnut}_{surf} &= \beta_{Nut} \cdot \mathrm{NUT}_{mobile,l=0} \cdot \mathrm{Q}_{surf} \\ \mathrm{Fnut}_{lat,l=0} &= \beta_{Nut} \cdot \mathrm{NUT}_{mobile,l=0} \cdot \mathrm{Q}_{lat,l=0} \\ \mathrm{Fnut}_{lat,l} &= \beta_{Nut} \cdot \mathrm{NUT}_{mobile,l} \cdot \mathrm{Q}_{lat,l} \end{aligned} \tag{47}$$


In deep soil layers, $\beta_{Nut}$ which is the nutrients percolation coefficient, it controls the amount of NUT removed from the surface layer in runoff relative to the amount removed via percolation Neitsch et al. (2002). The value for $\beta_{Nut}$ can range from 0.01 to 1.0. We choose for $\beta_{Nut}$ a value of 0.4 (similar to Von Bloh et al. (2018)).

Nutrients flux moved to the lower soil layer with percolation $\mathrm{Fnut}_{perc,l}$ is calculated in equation 48 as:

$$\mathrm{Fnut}_{perc,l} = \mathrm{NUT}_{mobile,l} \times \mathrm{w}_{perc,l} \tag{48}$$

$\mathrm{Fnut}_{perc,l}$ is a sink for the current NUT concentration in the soil layer and a source for the NUT pool of the soil layer just underneath (equation 49).

$$\mathrm{LEACH}_{\mathrm{NUT},l} = \begin{cases} \mathrm{Fnut}_{perc,l} - \mathrm{Fnut}_{surf} - \mathrm{Fnut}_{lat,l} & \text{for } l = 0 \\ \mathrm{Fnut}_{perc,l-1} - \mathrm{Fnut}_{perc,l} - \mathrm{Fnut}_{lat,l} & \text{for } l > 0 \end{cases} \tag{49}$$





## 4 Model set-up and inputs

We start by spin-up simulation, where water pool and carbon are initialized to zero during 5000 years to bring natural vegetation patterns, and carbon stocks into dynamic equilibrium. We cyclically repeat 30 years of climate data input with constant concentrations of atmospheric carbon dioxide (at 278 ppm). Then a second phase of spin-up (during 500 years) during which land use is introduced in the year 1700 from which it is updated annually according to the historic land use data-set (see Fader et al., 2015, 2010, and Sect. 4.1). And finally we run a hind cast simulation [1960-2000] with the new external input of nutrients described in the following section.

### 4.1 Model input

The model domain encompasses the catchments of rivers ending in the Mediterranean Sea (see Fig. 2). We must note that, since the assessment of some of the simulated variables requires data available at the administrative level, we run the model over the entire area of the Mediterranean countries, i.e. also for catchments not ending in the Mediterranean Sea. The required input data for LPJmL-Med are: (i) gridded climate variables (temperature, precipitation, and radiation); (ii) atmospheric $CO_2$ concentrations; (iii) gridded soil texture as described in Schaphoff et al. (2018); (iv) gridded land use and crop distribution dataset; and (v) for the present application we have implemented three new inputs quantifying the nutrients load from agricultural management and wastewater release (see below).

### Climate and soil inputs

The model runs on a daily time step. For the present study, we used monthly climate inputs data (*i.e.* precipitation and temperature data) from the regional climate model CNRM-ALADIN (Herrmann et al., 2011; Farda et al., 2010; Déqué and Somot, 2008) in the framework of Med-CORDEX programme (https://www.medcordex.eu/). Here we use the high-resolution version at 12 km (converted to a resolution of 5 by 5 arc-minutes) for the area covered by the domain of the CNRM-ALADIN model (*i.e.* between 27°N and 57°N in latitudes and between 10°W and 40°E in longitudes). Monthly temperature is linearly interpolated, while a generator based on monthly precipitations and monthly number of wet days provide daily precipitation values (Gerten et al., 2004). To complete the missing ares (*i.e.* southern 27°N) we have used the CNRM-CM5 global model (Voldoire et al., 2013); from the ESGF data set, https://esgf-data.dkrz.de/projects/esgf-dkrz/). Incident solar radiation is internally computed from the cloudiness data of the Climate Research Unit's (CRU) time series (TS) 3.1 data (Mitchell and Jones, 2005). LPJmL works with soil hydrological parameters that depend on soil texture. We use the 13 USDA soil texture classes that are provided by the Harmonized World Soil Database v 1.2 (FAO, 2012) at 30 arc-second spatial resolution. The original data have been agregated at 5 arc-minute resolution.

### 4.1.1 Land-use and river network data

The land use data for the crops in LPJmL-Med had been compiled from different sources (Portmann et al., 2010; Monfreda et al., 2008; Klein Goldewijk et al., 2011), as explained in Fader et al. (2015, 2010). Decadal cropland data from HYDE were





interpolated to derive annual values and then used for extrapolating the land use patterns of ∼2000 to the past, until 1700. Historical irrigation fractions were determined as explained in Fader et al. (2010). Further information is given by Fader et al. (2010, 2015).

Runoff is simulated by the model for each grid cell, and a drainage direction map gives the transport directions of the flows towards the different rivers. The river-routine scheme is derived from the Hydrosheds database (Lehner et al., 2008) and is updated at 5 arc minute resolution (Siderius et al., 2018). The GRanD database (Lehner et al., 2011) provides detailed information on water reservoirs that includes information on storage capacity, total area, and main purpose. Furthermore, information on natural lakes is obtained from Lehner and Döll (2004).

**Fertilizers, manure , and wastewater release**

**Fertilizer inputs**

Country-specific yearly N and P fertilizer consumption data are provided by the international fertilizer association (IFA, 2016) since 1961. The LPJmL simulations running within the 2010 administrative country boundaries, adjustments are done for countries where boundaries have changed (*e.g.* in eastern Europe after the end of the cold war). These adjustments are detailed in Appendix B. Each year and in each country, the amount of fertilizer is distributed over the grid cells according to their

crop fractions (with the exclusion of legumes). Our crop fertilization calendar considers two or three application dates for the different crops (*cf.* Table 1). Half of the N fertilizers are provided as nitrate $NO_3$, and half as ammonium $NH_4$. Maps illustrating the temporal dynamics of fertilizer inputs over the crop areas in the Mediterranean countries are shown in Fig. 3. The equations used for calculating the annual nitrogen and phosphorus content of Fertilizer are described in Section 2.3.1.

**Manure inputs**

The Food and Agriculture Organization provides country-specific yearly data on N manure applied to cultivated soils or left on pasture (FAO, 2016a) since 1961. As for fertilizers, adjustments are done when country boundaries have changed (*cf.* Appendix B). N:P ratios in manure derived from Potter et al. (2010, 2011a) are used for estimating P manure applied to cultivated soils or left on pasture. Unlike fertilizers providing mineral N and P nutrients, manure provides organic nutrients that are not immediately available for the plants, therefore manure is applied earlier in the year than the synthetic fertilizers.

Our crop fertilization calendar considers two annual application dates for all crop categories (*cf.* Table 1). For pasture, manure is equally distributed between May 1st and September $30^{th}$. Maps illustrating the temporal dynamic of manure inputs over the pasture and crop areas in the Mediterranean countries is shown in Fig. 4. The equations used for calculating the annual nitrogen and phosphorus content of Manure are described in Section 2.3.2.

**Wastewater inputs**

Wastewater nutrient content depends on population connected to public sewerage system and on country-specific Gross Domestic Product (GDP). High GDP generally implies a high level of households owning washing machines and dishwashers,



therefore a high consumption of detergents (containing more or less pollutants depending on regulations), but it is also associated with a more widespread access to sewerage systems with wastewater treatment. Thus, depending on the demography dynamics and on the GDP change, the wastewater inputs can increase or decrease. The equations and data used for calculat-
ing the annual nitrogen and phosphorus content of wastewater are described in Section 2.3.3. Maps illustrating the temporal dynamics of wastewater inputs over the Mediterranean countries is shown in Fig. 5.

## 4.2    Model outputs

As a function of agricultural management and climatic conditions, LPJmL-Med simulates, spatially explicitly and at a daily to yearly temporal resolution, growing periods (sowing and harvest dates), gross and net primary productivity, carbon stock
in plants, litter, and soil, agricultural production, as well as a number of hydrological variables, such as soil evaporation, infiltration, percolation, water stress, irrigation water requirements, runoff, and river discharge. In the frame of the present study we have added a new output to the LPJmL-Med model: nutrient concentrations ($PO_4$ and $NH_4$, and $NO_3$) in surface and deep soil, in lake water, and in water discharge, as well as phytoplankton content in lake and in water discharge.

## 5    Results and discussion

In the following section, the capacity of LPJmL-Med for simulating the water discharge and nutrient concentrations was compared to published in-situ data, with a focus on the main rivers of the Mediterranean Sea. In general, data coverage of the northern-European rivers is good, whereas data on southern-Mediterranean rivers is poorer. Particularly, long time series of data are missing for southern-Mediterranean rivers.

### 5.1    Water discharges from the main rivers of the Mediterranean Sea

Water discharge is the main factor controlling nutrient transfer from land to sea by rivers. We therefore start our assessment of LPJmL-Med with an evaluation of the rivers water discharge to the Mediterranean. Moreover, the freshwater discharges into the sea can have an important influence on the functioning of marine ecosystems through their control of the ocean dynamics in the Mediterranean Sea (e.g. Skliris et al., 2007). For all these reasons, it is essential to accurately simulate river discharge. This is particularly true in the Mediterranean context, where water resources represent an important economic value and political
barriers probably still inhibit greater transparency regarding river data. We chose for the comparison the period between 1920 and 1985 because data for the main rivers (*i.e.*, Rhone, Po, Ebro, but not Nile) are available for this period, and to limit the impact of damming and anthropogenic water use.

Figure 6 shows the water discharge time series for the Rhone, Po, Ebro and Nile rivers against historical in-situ data between 1920 and 1985. The LPJmL-Med simulates relatively well the interannual variability of the main rivers in the Mediterranean
Sea especially the Po, Rhone and Ebro. A very high correlation was found between model outputs and in-situ data (Fig. 7), *i.e.* the R-square value is higher than 0.95 for the three rivers, showing that the interannual variability is clearly well simulated by the LPJmL-Med model (Fig. 6 and Fig. 7). Almost all of them presented a significant discharge decrease as a result of the





combined effect of ongoing climate change and enhanced anthropogenic water use (Ludwig et al., 2009, 2003). The Ebro river show a net decrease of water discharge between 1960 and 1990 (from 28 to 9 $km^3.y^{-1}$) due to the higher frequency of dry periods observed in the Ebro River basin during the same period (Valencia et al., 2010). However, the general trend for the Po and the Rhone rivers is to remain about constant for the same period. This distinctive behaviour pattern may be explained by the non-Mediterranean climate in the upper northern parts of their basins (Lutz et al., 2016; Ludwig et al., 2009). The succession of dry and wet periods that can be seen on in-situ data for the three rivers is well reproduced by the model, indicating that the seasonal variability of water discharge is simulated by the LPJmL-Med. However, as shown in Figure 8, the amplitude of the seasonal cycle is larger in the model because we do not represent explicitly the role of dams in the regulation of water flows (*i.e.*, release and retention of water). The construction of dams in the context of water resources management plays an important role matching human needs with the hydrological regime, *i.e.*, to ensure an adequate supply of water by storing water in times of surplus and releasing it in times of scarcity (Fig. 8). This has been well studied for the Ebro catchment (Radinger et al., 2018). It is difficult to represent explicitly this water regulation by dams because very few data are available and those regulations are sporadic and hard to characterize, particularly for long periods.

It is also very difficult to evaluate and simulate the discharge from the Nile, which loses most of its water through infiltration in swamps, river evaporation and anthropogenic water use (Nixon, 2003). Since 1965, the construction of the Aswan High Dam has had a major impact on the water discharge. The flow rate of the Nile river is between 6 $km^3yr.^{-1}$ (ElElla, 1993) and 15 $km^3.yr^{-1}$ (Nixon, 2003). In another study, Skliris et al. (2007) have estimated the flow of the natural long-term discharge at Aswan at about 83 $km^3.yr^{-1}$. However, the Aswan dam is still very far from the Mediterranean Sea and the anthropogenic water use in this area is very low compared to the northern part of Egypt in the Nile delta. In this context, LPJmL-Med simulates relatively well the water discharge of Nile with an average value of 17.88 $km^3.yr.^{-1}$ (average value between 1965 and 1985), despite being relatively higher than the previous estimation from literature (ElElla, 1993; Nixon, 2003), but still far lower than the estimation of Skliris et al. (2007) of about 83 $km^3.yr^{-1}$ at Aswan.

## 5.2 Spatio-temporal variability of nutrients

### 5.2.1 Nitrate (NO$_3$)

Figure 9 shows the NO$_3$ (in $kt.y^{-1}$ of N-NO$_3$, noted here NO$_3$) discharge simulated by LPJm-Med for the main rivers flowing into the Mediterranean Sea. NO$_3$ fluxes in the rivers Rhone (Fig.9a), Po (Fig. 9b) and Ebro (Fig.9c) rivers increased steadily from the beginning of the 1960s up to the 1990s. The model simulates a rapid increase in NO$_3$ in recent years after 1994, with a greater interannual variability for the three rivers. The same increase in NO$_3$ was presented in Ludwig et al. (2009, 2003) combining in-situ data and the NEWS-DIN model of Dumont et al. (2005), and also from the MOOSE (Mediterranean ocean observing system on environment) observatory system (Raimbault and Lagadec, 2012) available between 1980 and 2004 for the Rhone River. The NO$_3$ increase could be explained by the additional inputs from human activities which often dominate the natural sources, and it has been postulated that between 1970 and 1990, humans increased the global delivery of dissolved inorganic N to the oceans by a factor of three (Smith et al., 2003), as a consequence of increased consumption of nitrogen





fertilizers from 1960 to 1990 by a factor three (from 200 kt.y$^{-1}$ to more then 600 kt.y$^{-1}$) for the watersheds of the main rivers (*cf.* Fig. 3). In addition high application rates of manure are found in the northern Mediterranean drainage basins, such as those of the Ebro and Rhone rivers (*cf* Fig. 4). However, there is an inconsistency between model outputs and observations for recent years, where Ludwig et al. (2009) and MOOSE NO$_3$ data remained approximately constant, or even decreased slightly, while

LPJmL-Med simulates higher values for NO$_3$, especially for the Ebro River: in plain terms, the Ludwig et al data show NO$_3$ discharge at around 40 kt.y$^{-1}$ at the end of 1990s, while the LPJmL-Med simulates more then 100 kt.y$^{-1}$ for the same period. The rapid increase in NO$_3$ discharge in the model is clearly associated with the high NO$_3$ concentration in fertilizer and manure over the watershed of the Ebro, as a consequence of the impressive growth of agrarian production in Spain, multiplied by 3.33 between 1900 and 2008 (Molina et al., 2016).

For the Nile river (Fig. 9d) where nitrate data are missing, LPJmL-Med simulates an increasing discharge of NO$_3$ from 1970, certainly due to the huge amount of NO$_3$ in the inputs data (*cf.* section 4) due to the demographic explosion in Egypt and to changes in agricultural practices. In addition the Nile River has one of the highest average nitrate concentrations (Ludwig et al., 2003), although this value has only been derived from a few published values, and it is not clear whether this value is really representative for this river (for more detail see Ludwig et al., 2003).

### 5.2.2 Phosphate (PO$_4$)

Phosphate (in kt.y$^{-1}$ of P-PO$_4$ noted here PO$_4$) output trends are more heterogeneous (Fig. 10). There is a strong increase in the PO$_4$ discharge simulated for the four main Mediterranean rivers at the beginning of the 1960s. PO$_4$ discharges reached their maximum value between 1975 and 1980 for the three rivers (Rhone, Po, and Ebro), and relatively later for the Nile river. This evolution stops after the beginning of 1980s, and the value started to decrease continuously, but the pattern of change is

not completely in phase between the main rivers, since the beginning of the phosphate decline differs between the three. It started earlier in the Rhone and Ebro (about 1980), followed by the Po (around 1985), and finally by the Nile (after 1990). This difference reflects the time lag between the different countries in the regulation of phosphorus pollution through the banning of phosphorus detergents and the systematic implantation of wastewater treatment plants. At the end of the 1990s the phosphate discharges again reach the values they had at the beginning of the 1970s'. This indicates that the upgrading of wastewater

treatment has been successful. Phosphorous load from industry has also been reduced due to the use of cleaner technology.

The comparison with the observed PO$_4$ from Ludwig et al. (2003, 2009), and from MOOSE data-set (Raimbault and Lagadec 2012) shows a good agreement with simulated PO$_4$, especially for the Rhone River (Fig. 10b), and relatively higher then the data provided within the Moose observation network for the period between 1980 and 2004. However, LPJmL-Med simulates higher values than Ludwig et al. (2009) data for the Po River, after a slight decrease around 1973 due to sharp decrease in PO$_4$

at the beginning of 1970 in wastewater (cf. Fig. 5d) followed by a rapid increase until 14 kt.y$^{-1}$ in the model compared to 8 kt.y$^{-1}$ in Ludwig et al. (2009) at the beginning of 1980. On the other hand, the LPJmL-Med model simulates a lower PO$_4$ discharge than that from the Ebro River. In the case of the Pô River, it is likely that the high simulated values are associated with the high level of PO4 in fertilizers (*cf.* Fig. 3d), and in wastewater (cf. Fig.5d). In contrast, PO$_4$ concentrations are very low in fertilizers and wastewater input for the Ebro River (*cf.* Fig.3d and Fig. 5d).





Again, the patterns of change in phosphate in the Nile River are different compared to the other large rivers. Here, the curve in Fig.10e does not show a convex shape like in the three other rivers. A very slow decline of $PO_4$ is simulated in the Nile River after 1990 as compared to the Europeans countries, thereby indicating that the progression to better water quality has been more rapid in western Europe than in southern countries.

### 5.2.3   Comparison of P and N trends in the Mediterranean Sea

Figure 11 shows the $NO_3$ and $PO_4$ inputs to the eastern basin (EMed) and western basin (WMed) through river discharge and water runoff transported across the whole basin in the LPJmL-Med grid (*cf.* Fig. 2). It can first be seen that both $NO_3$ and $PO_4$ in water discharges are much higher in the EMed than in the WMed. The differences between WMed and EMed are clearly associated with the particularly high nutrient concentrations in the Nile and Po rivers, which have a considerable impact on the budgets of the entire eastern Mediterranean. Comparison with the in-situ data from Ludwig et al. (2009) suggests a good

agreement with the LPJmL-Med outputs for the two basins, the difference between the EMed and the WMed being quite well reproduced by the model with a higher discharge in the EMed (Fig. 11). The huge increase in $PO_4$ discharge at the beginning of 1960s (from 20 to 80 kt.y$^{-1}$ in the WMed and from 55 to 160 kt.y$^{-1}$ in the EMed) is particularly well simulated by the model, as well as the $PO_4$ drop at the end of the 1980s (Fig. 11b). This huge decrease reflects the adoption of new legislation by the surrounding countries, such as the prohibition of phosphorus detergents, and the improvement of wastewater treatment plants,

as well as other features (see below). By contrast, $NO_3$ discharge shows a constant increase over the whole period (especially in the WMed) which is well reproduced by the model (Fig. 11a), but as already analyzed for the rivers Rhone, Po and Ebro, the model simulates higher $NO_3$ concentrations during the recent period, particularly for the Ebro River (Fig. 11a, Fig. 9).

The patterns of change in the river fluxes of N and P exhibit opposite trends in the Mediterranean Sea after the 1980s. Both enhanced by anthropogenic activities in the drainage basins at the beginning of the 1960s, especially P with a dramatic increase

in the 1960s and 1970s. However, the efforts undertaken to mitigate point source pollution at the beginning of the 1980s had an immediate impact on the $PO_4$ loads from rivers, in particular European rivers, where the mean annual fluxes decreased from the second half of the 1980's. This pattern of change in P is mainly due to the improvement of the water quality, with the introduction of tertiary treatment (*i.e.* with phosphorous removal) and to the banning and abandon of phosphate detergents at the beginning of 1980 especially in Europe. Crouzet et al. (1999) have estimated that between 50 and 75% of the dissolved

phosphorous is derived from point sources (*i.e.* urban wastewater), which confirms the stronger dependence of phosphorous loads on point source pollution, while the other sources (*i.e.* agriculture) generally account for 20 to 40% (Ludwig et al., 2009). These trends are in good agreement with the the changes in $PO_4$ published in Ludwig et al. (2003, 2009). On the other hand, the pattern is different for $NO_3$, which followed a more or less a continuous increase from the beginning of the 1960s until the present in all three rivers. N input is usually dominated by diffuse sources such as fertilizers from agriculture (Ludwig et al.,

2003). Crouzet et al. (1999) have estimated that about 45-90% of the nitrogen load in inland waters is related to agriculture in Europe. Furthermore, wastewater and deposition of nitrogen oxides (NOx) in land and surface water may also have contributed to increased N during the recent years (Ludwig et al., 2009).



Finally, a quantitative comparison between in-situ data and model outputs has also been undertaken using a Taylor diagram (Taylor, 2001) that shows the standard deviation, the correlation between the data and the model and the root mean square error for water discharge (Fig. 12a), $NO_3$ (Fig. 12b) and $PO_4$ (Fig. 12c). For water discharge, correlations for the three rivers (Rhone, Po, Ebro) are higher than 0.94, with relative standard deviations ranging between 0.7 and 0.8, showing that the LPJmL-Med can predict relatively well the water discharge for the main rivers in the Mediterranean Sea for the historical period (between 1920 and 1980).

For nutrients, the correlation with data is less favorable, and certainly some aspects in the simulation still need to be improved, regarding both the representation of processes and input data (see below). The poor correlation could also be partly ascribed to the limited number of available long time series of in-situ data, which is well below the number of data for the water discharge. The only long-term data presented in Ludwig et al. (2003, 2009) are produced by the combination of in-situ data and the NEWS-DIN model, which makes more complicated the development of modelling approaches and their evaluation. In the case of the Rhone river, the MOOSE observatory system provides regular monitoring of nutrient fluxes in the Rhone. However, in-situ data from the MOOSE program show lower concentrations during the peak of $PO_4$ in the 1980s as compared to data from Ludwig et al. (2003) (*i.e* $\sim 9$ kt.y$^{-1}$ in Ludwig et al. (2003) and $\sim 6$ kt.y$^{-1}$ from MOOSE program). The monitoring programs are generally based on monthly/yearly sampling, and therefore not appropriate to quantify the short and violent flash-floods which are typical for the hydrological regimes in the Mediterranean climate (Estrela et al., 2000). If these events are not monitored by suitable sampling strategies, it is not possible to assess the average fluxes, and the ratio of peak discharge to mean annual discharge is still very high. This is particularly true for the long period for the small Mediterranean rivers (Estrela et al., 2000). Furthermore, the Mediterranean river basins are highly reactive to local climatic features, they may be more vulnerable in the context of climate change and alterations in the frequency of extreme climatic events (floods, droughts) can have a severe impact on the river fluxes.

Moreover, the module to simulate nutrient discharge was implemented for the first time in LPJmL-Med model, and it was tested for the first time in this work. Several mismatches between model and observations could be associated with our approach as described above (see Sect. 2), but also to the input data (of fertilizer, wastewater, manure, and land-use patterns used in this study) described in section 2 and 3. In particular, the IFA input time-series considers only national data. Gridded data sets exist, but they are available for specific periods only. Their combination with yearly gridded crop areas and country-level and (when possible) sub-national fertilizer data could enable the establishment of fertilizer input data that better reproduce the spatio-temporal dynamics which could allow a better representation of the spatial differences in fertilizers inputs. The same holds true for manure inputs. Nevertheless, all these comparisons (Fig.9, Fig.10, Fig.11, and Fig.12) led us to conclude that LPJmL-Med was able to reproduce most of the major nutrients (*i.e.* $NO_3$ and $PO_4$) features and this provided a basis for realistic values for water and nutrient discharge of the major river into the Mediterranean Sea, as well as at regional scale for the two basins, with a significant correlation and a relatively high R-square for the two basins (ranging between 0.7 and 0.8 for $NO_3$, and between 0.6 and 0.7 for $PO_4$, Fig. 12). These results suggest that this approach is appropriate for generating a simulation that is sufficiently realistic on decadal timescales, and could be used to investigate the effects of the variations of river discharge in the Mediterranean Sea on marine ecosystems. However, some aspects in the simulation still need to be improved





especially the inconsistency between model and in-situ data of $NO_3$ for recent years: where data remained approximately
constant, the model simulates à rapid increase of $NO_3$ for almost all the rivers. An improved representation of some of the

involved processes in future work could improve the simulation of $NO_3$ in river flow (note that very few in-situ data of NH4
are available, that hardly allows us to evaluate the model performance for this state variable). One of the main limitations of
the present study is the number of available in-situ data covering the whole Mediterranean basin for input forcing data (*i.e.*
fertilizer, manure, wastewater release data) and for nutrient concentrations in runoff and in river discharge. Therefore, it will
be necessary to incorporate new in-situ data (as previously explained), or to develop new statistical approaches to validate our

modeling approach in future work.

## 6    Conclusions

This study proposes the first basin-wide simulation at 1/12° of water discharge and nutrient release (N and P) into the Mediter-
ranean Sea through the implementation of the biogeochemical land-sea nutrient transfer processes within the agro-ecosystem
model LPJmL-Med. For this purpose, the representation of the nutrient transfer from land to sea has been introduced into

LPJmL-Med by considering the following processes: remineralization, denitrification, adsorption, nitrification, and phyto-
plankton dynamics, and a compilation of a new input data set of fertilizer, manure and wastewater nutrients content has been
added to the model.

The model successfully simulates the interannual variability of water discharge for the main rivers in the Mediterranean Sea
especially the rivers Po, Rhone and Ebro, where we find a very high correlation (the R-square values are higher than 0.94 for

the three rivers). The Ebro river show a net decrease of water discharge between 1960 and 1990 (from 28 to 9 $km^3.y^{-1}$) due
to the higher frequency of dry periods observed in the Ebro River basin during the same period. However, the general trend for
the Po and the Rhone rivers remains about constant for the same period; this distinctive behaviour pattern may be explained
by the non-Mediterranean climate in the upper northern parts of their basins, which results in a high precipitation load. The
succession of dry and wet periods is well simulated by the model relative to in-situ data for the three rivers, indicating that the

seasonal variability in water discharge is well simulated by the LPJmL-Med. However, the amplitude of the seasonal cycle is
greater then observations because we do not represent explicitly the role of dams in the regulation of water flows (*i.e.*, release
and retention of water). It is very hard to evaluate and simulate the discharge from the Nile, which loses most of its water
through infiltration in swamps, river evaporation and anthropogenic water use. and the construction of the Aswan High Dam
in 1965 had a major impact on the water discharge.

Results show a good consistency between the simulated nutrient concentrations ($NO_3$ and $PO_4$) and available in-situ data.
The patterns of change in river fluxes of $NO_3$ and $PO_4$ exhibits opposite trends in the Mediterranean sea. $NO_3$ followed a
more or less continuous increase from the beginning of the 1960s until the present in all three rivers. $PO_4$ trends are more
heterogeneous. There is a strong increase of $PO_4$ between 1960 and 1980, after that the mean annual fluxes declined from
second half of the 1980s as a consequence of the banning of phosphorus detergents, and the development of waste water

treatment plants in the different countries. The freshwater fluxes do not control the trends of nutrients; in particular, the strong





NO$_3$ increase in the Ebro is completely decoupled from the significant discharge decreases as a consequence of massive dam constructions, as well as in the Rhone and the Po where NO$_3$ concentration increase while the trend for water discharge has been almost at a steady state for the last decades. Furthermore, the two nutrients do not have the same origin in the Mediterranean region: P concentrations are mainly controlled by wastewater release, while those of N result from agriculture (*i.e.* fertilizer and manure).


**Table 1.** Crop fertilization calendar. FRAC$_{fert}$ in column 3 corresponds to the fraction of total manure or inorganic fertilizers applied on each application day. Note that FRAC$_{fert}$ =0 the rest of the year. [a] Weather condition: no rain on the day of the application and on the 2 previous ones

| Crop type | Fertilizers or manure? | FRAC$_{fert}$ | Application 1 (A1) | Application 2 (A2) | Application 3 (A3) |
|---|---|---|---|---|---|
| Winter crops | Fertilizers | 1/3 | At sowing | From the 1st of April[a] | From 3 weeks after A2 |
| | Manure | 1/2 | At sowing | From the 1st of March[a] | no |
| Summer crops | Fertilizers | 1/2 | At sowing | From 3 weeks after A1[a] | no |
| | Manure | 1/2 | At sowing | From 3 weeks after A1[a] | no |
| Permanent crops | Fertilizers | 1/2 | From the 1st of April[a] | From 3 weeks after A1[a] | no |
| | Manure | 1/2 | From the 1st of March[a] | From 3 weeks after A1[a] | no |
| Pastures | Manure | 1/162 (North) 1/365 (South) | Uniformly applied over the grazing season (April to September in the northern hemisphere, all year long in the southern hemisphere) | | |

*Code availability.* The original code of the LPJmL model is publicly available through PIK's gitlab server at https://gitlab.pik-potsdam.de/lpjml/LPJmL. The source code of the adjusted LPJmL-Med version developed in this study and described here are available through osupytheas's gitlab server at https://gitlab.osupytheas.fr/mayache/lpjml-med_version1 The output data from the model simulations described in this study can be obtained from the corresponding authoron reasonable request, and should be referenced as Ayache et al.

*Data availability.* The data associated with the paper are available from the corresponding author upon request. All the data used in this study were published by their authors as cited in the paper. Here we present the model result against the in situ data already published in the literature.



**(a)**

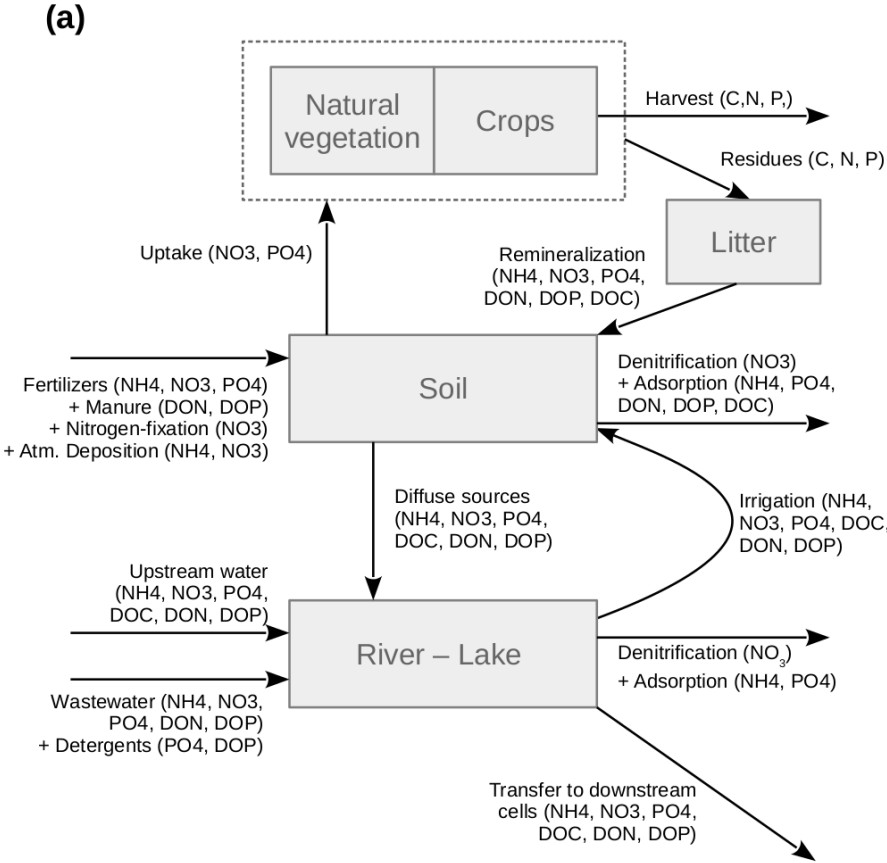

**(b)**

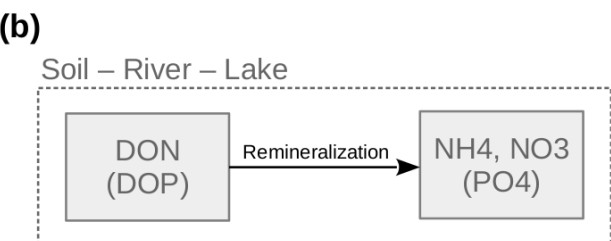

**(c)**

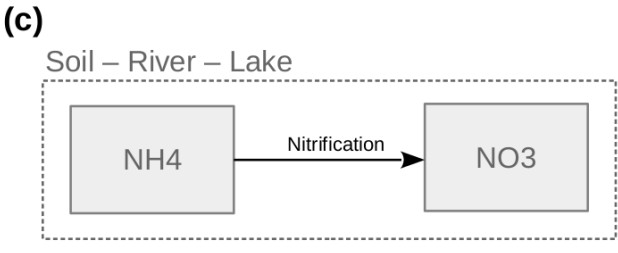

**Figure 1. (a)** Transfer of nutrients in a gridcell. **(b -c)** Transformation of nutrients occuring in soils and water.



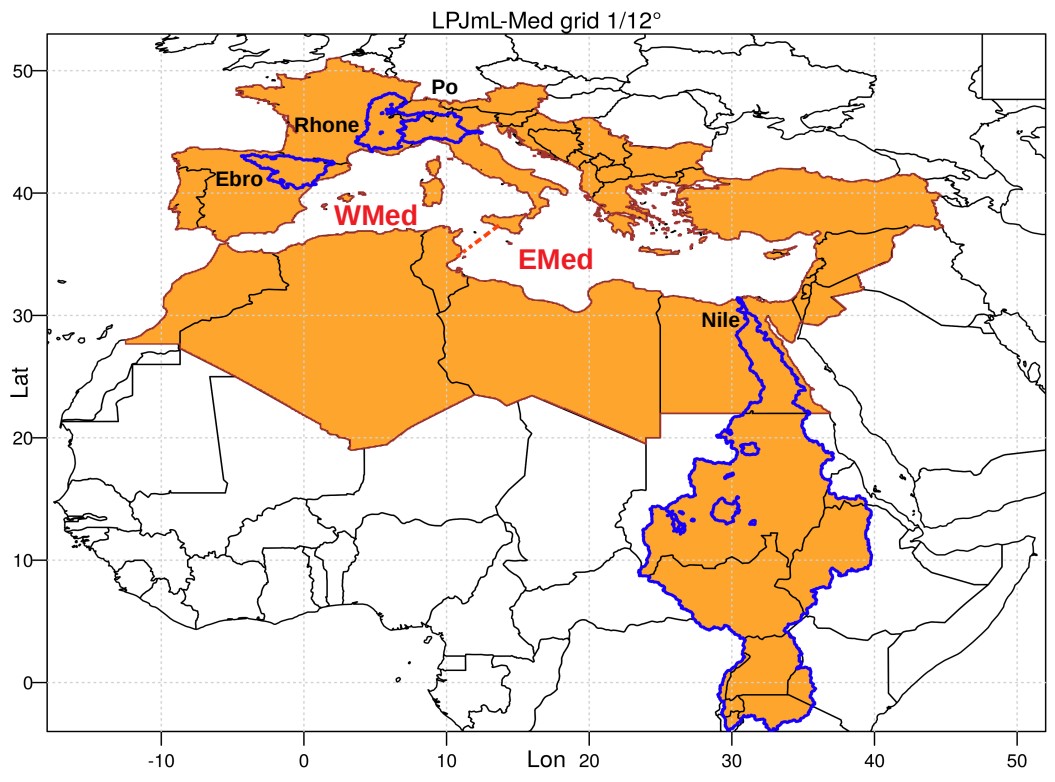

**Figure 2.** Map of theLPJmL-Med model domain (1/12 degree resolution). It includes all catchments of rivers flowing into the Mediterranean, plus the whole area of Mediterranean or related countries (see text). In blue bold the watershed of the Nile, Ebro, Po and Rhone rivers. Red line present the frontier between the western (WMed) and eastern (EMed) Mediterranean basins as presented in Millot and Taupier-Letage (2005).



**Figure 3.** A compilation of nutrient inputs data set from fertilizer data (in kilotons/pixel of NO3 and PO4), (**a** and **b**) horizontal maps represent LPJmL-Med annual input of fertilizers averaged between 1960 and 2005. (**c** and **d**) time series for Rhone catchment in red, Po catchment in blue and Ebro catchment in green. (**e** and **f**) times series for eastern and western basins.







**Figure 4.** Same as Fig. 3 but for Manure data in kilotons of NO3 and PO4







**Figure 5.** Same as Fig. 3 but for wastewater release (in tons/pixel of NO3 and PO4)





**Figure 6.** Yearly average water discharge in m³/s for the main rivers into the Mediterranean Sea. **a**) horizontal map on implemented rivers in the LPJmL-Med grid (Averaged between 1963 and 2000). **b**, **c**, **d**, and **e** time series of water discharge at the mouths of Rhone, Po, Ebro and Nile, successively. Model output in blue, in-situ data (from the Global River Discharge database RivDIS Vörösmarty et al. (1996) in red



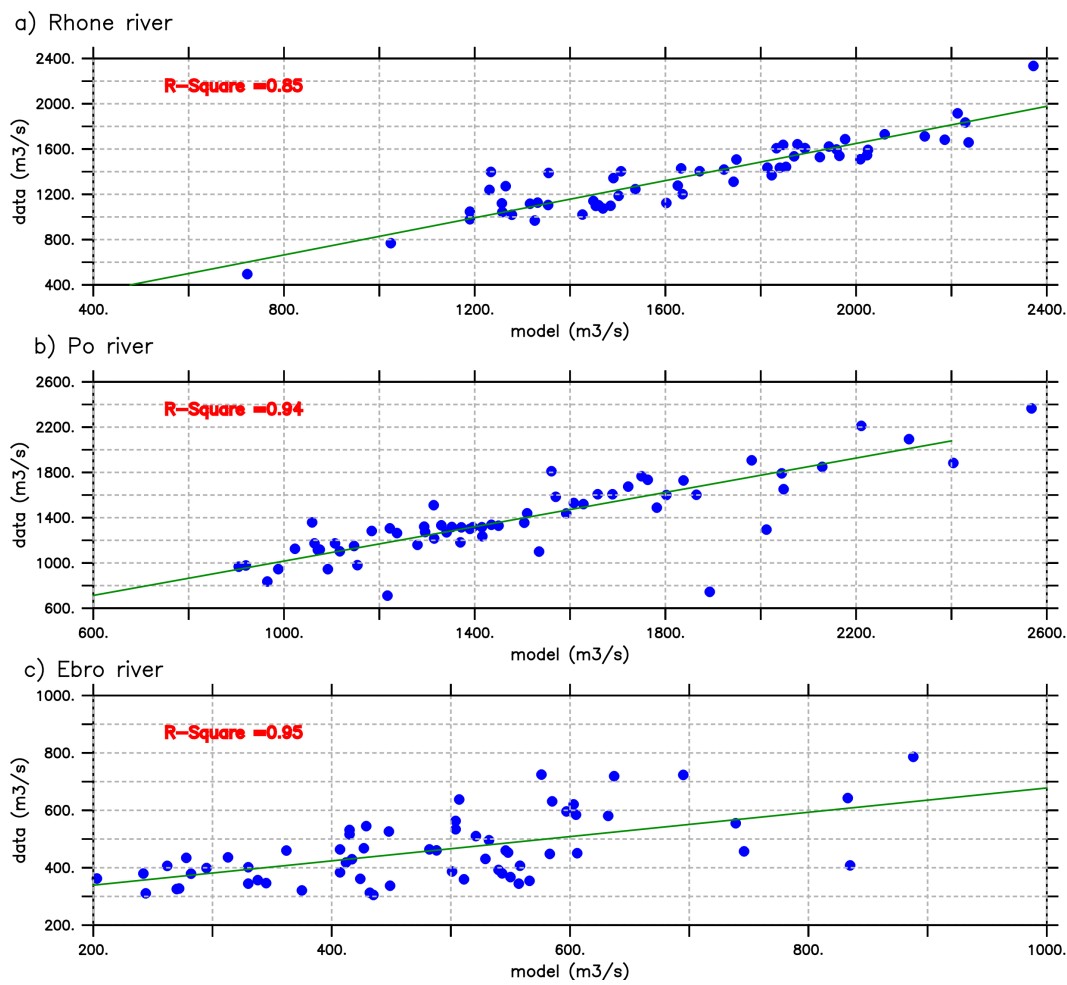

**Figure 7.** Scatterplot of LPJmL-Med simulated water discharge in m3/s (averages over 1920-1980) versus in-situ data from the Global River Discharge database RivDIS (Vörösmarty et al., 1996) at the mouths of the **a)** Rhone, **b)** Po and **c)** Ebro rivers.



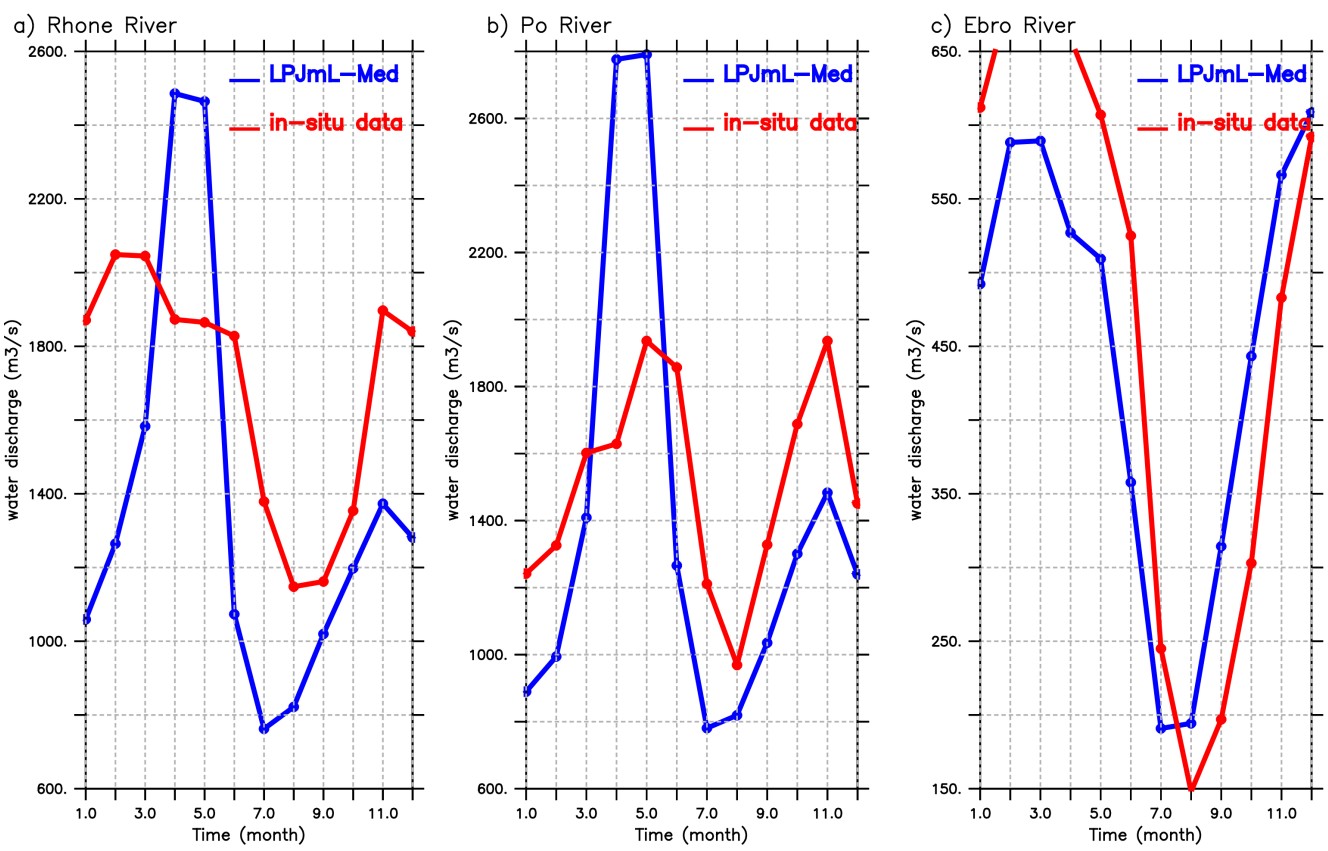

**Figure 8.** Simulated and observed seasonal cycle of monthly average water discharge in m$^3$/s for **a)** Rhone, **b)** Po, and **c)** Ebro rivers. In-situ data from the Global River Discharge database RivDIS (Vörösmarty et al., 1996) (averaged over 1920-1980).





**Figure 9.** Annual NO₃ simulated by LPJm-Med in kt.y⁻¹ flowing from the mains rivers into the Mediterranean Sea. **a**) horizontal map averaged between 1963 and 2000. (**b, c, d**, and **e** time series for the Rhone, Po, Ebro, and Nile rivers successively. Model output in blue, in-situ data (from Ludwig et al. 2009) in red, and in green from MOOSE program from Raimbault and Lagadec (2012)



**Figure 10.** same as Fig. 9 but for PO$_4$.





**Figure 11.** $NO_3$ **(a)** and P-$PO_4$ **(b)** inputs from rivers and runoff in kt.$y^{-1}$ for the western basin in blue and for the eastern basin in red from the LPJmL-Med outputs (plain line) and from Ludwig et al. (2009) data (dotted line).



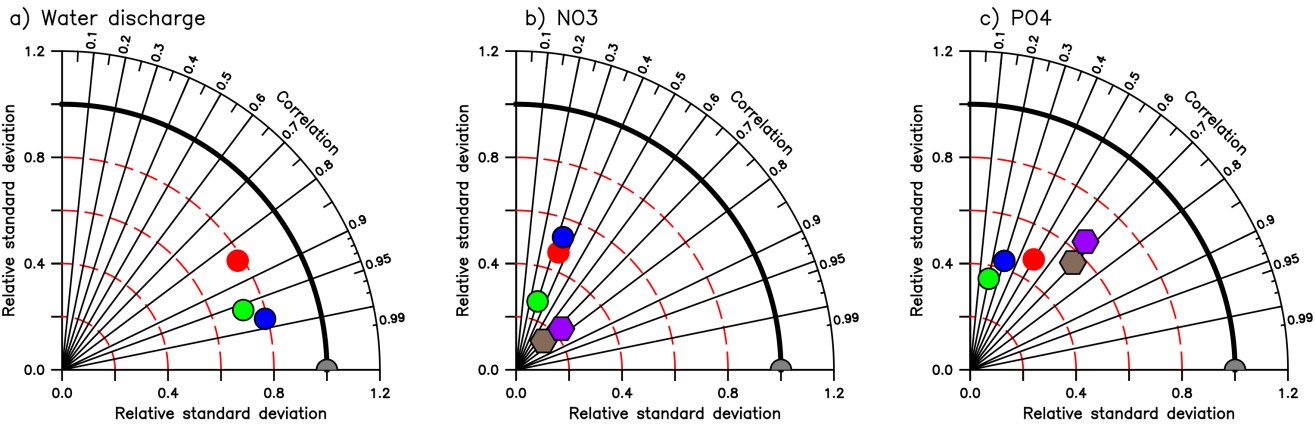

**Figure 12.** Taylor diagram showing the standard deviation of the normalized model outputs, the correlation between in situ data and model outputs for the Rhone River (in red), the Po River (in blue) and the Ebro River (in green), western basin (in brown), and eastern basin (in purple) . **a)** water discharge (data from Vörösmarty et al. 1996 betwwen 1920 and 1985), **b)** NO$_3$ concentrations (data from Ludwig et al. 2009, between 1960 and 2000), **c)** PO$_4$ concentrations (data from Ludwig et al. 2009, between 1960 and 2000).





Table A1: Value and unit of global parameters used in the model from the literature. (a) depends on crop type (see Table 1), (b) depends on residue management practices.

| Symbol | Description | Unit | Value | References |
|---|---|---|---|---|
| **Wastewater release and treatment** | | | | |
| $\text{REM}_{N,0}$ | frac. of N removed from wastewater without treatment | - | 0 | |
| $\text{REM}_{P,0}$ | frac. of P removed from wastewater without treat. | - | 0 | |
| $\text{REM}_{N,1}$ | frac. of N removed with primary/mech. treat. | - | 0.1 | Van Drecht et al. (2003, 2009) |
| $\text{REM}_{P,1}$ | frac. of P removed with primary/mech. treat. | - | 0.1 | Van Drecht et al. (2009) |
| $\text{REM}_{N,2}$ | frac. of N removed with secondary/bio. treat. | - | 0.35 | Van Drecht et al. (2003, 2009) |
| $\text{REM}_{P,2}$ | frac. of P removed with secondary/bio. treat. | - | 0.45 | Van Drecht et al. (2009) |
| $\text{REM}_{N,3}$ | fraction of N removed with tertiary/adv treat. | - | 0.80 | Van Drecht et al. (2003, 2009) |
| $\text{REM}_{P,3}$ | fraction of P removed with tertiary/adv treat. | - | 0.90 | Van Drecht et al. (2009) |
| $\text{RATIO}_{N:P,ww}$ | N:P ratio of municipal wastewater | - | 6 | Van Drecht et al. (2009) |
| $\text{CONT}_{P,Ldet}$ | P content of laundry detergents | gP/g | 0.0625 | Van Drecht et al. (2009) |
| $\text{CONT}_{P,Ddet}$ | P content of dishwasher detergents | gP/g | 0.1 | Deloitte (2014) |
| $\text{CONS}_{hh,Ddet}$ | Household consumption of dishwasher detergents for dishwasher owning household | g/hh/d | 7 | Deloitte (2014) |
| PPHH | Average number of persons by household | pers/hh | 2.48 | Eurostat (2016) |
| **Nitrification and denitrification in soils and rivers** | | | | |
| $\text{VMAX}_{soil,nitr}$ | Maximum fraction of $NH_4$ nitrified daily in soil | $d^{-1}$ | 0.1 | Von Bloh et al. (2018) |
| $\text{VMAX}_{water,nitr}$ | Maximal daily nitrification rate in water | $\text{gN m}^{-3}\ \text{d}^{-1}$ | 0.72 | Billen et al. (1994) |
| $\text{K}_{water,nitr}$ | Half saturation constant for nitrification in water | $\text{g/m}^3$ | 1.4 | Billen et al. (1994) |
| $\text{VIT}_{water,den}$ | Uptake velocity for denitrification in water | m/d | $9.6\ 10^{-3}$ | Wollheim et al. (2008) Nevison et al. (2016) |
| $\text{LOAD}_{water}$ | Hydraulic load | m/d | 86400 | Hypothesis of 1m/s for the river routing |
| **Fertilizer application** | | | | |
| $\text{FRAC}_{fert}$ | Daily fraction of the annual application of fertilizers or manure; determined by the crop fertilization calendar | - | (a) | |
| **Adsorption in soils and rivers** | | | | |





| Symbol | Description | Unit | Value | References |
|---|---|---|---|---|
| $a$ | fixed parameter corresponding to the amount of P adsorbed for a concentration of 1 g/m$^3$ | gP/kg$_{soil}$ | $98.61 \cdot 10^{-3}$ | Mehdi and Sarfraz (2007) |
| $b$ | ffixed parameter corresponding to the buffer capacity for a concentration of 1 g/m$^3$ | m$^3$/kg$_{soil}$ | $50.76 \cdot 10^{-3}$ | Mehdi and Sarfraz (2007) |
| VMAX$_{PO4,ads}$ | Max. P adsorption rate on suspended particles in water | gP/g/d | 0.0055 | Billen et al. (2007) |
| Kads$_{water,PO4}$ | Half saturation constant for PO$_4$ adsorption in water | g/m$^3$ | 0.7 | Billen et al. (2007) |
| C$_{susp,forest}$ | Conc. of suspended solids in the water released from forest | g/m$^3$ | 50 | Billen et al. (2007) |
| C$_{susp,grass}$ | Conc. of suspended solids in the water released from grassland | g/m$^3$ | 70 | Billen et al. (2007) |
| C$_{susp,crops}$ | Conc. of suspended solids in the water released from crops | g/m$^3$ | 350 | Billen et al. (2007) |
| C$_{susp,urban}$ | Concentration of suspended solids in the water released from urban areas | g/m$^3$ | 500 | Billen et al. (2007) |
| **Nutrient uptake, remineralization and immobilization in soils** | | | | |
| RATIO$_{C:N,root}$ | C:N ratio of roots | gC/gN | 85 | Mooshammer et al. (2014) |
| RATIO$_{C:N,sap}$ | C:N ratio of sapwood | gC/gN | 300 | Mooshammer et al. (2014) |
| RATIO$_{C:N,leaf}$ | C:N ratio of leaves | gC/gN | 65 | Mooshammer et al. (2014) |
| RATIO$_{C:N,stor}$ | C:N ratio of storage | gC/gN | 65 | Mooshammer et al. (2014) |
| RATIO$_{C:P,root}$ | C:P ratio of roots | gC/gP | 1600 | Mooshammer et al. (2014) |
| RATIO$_{C:P,sap}$ | C:P ratio of sapwood | gC/gP | 5500 | Mooshammer et al. (2014) |
| RATIO$_{C:P,leaf}$ | C:P ratio of leaves | gC/gP | 1200 | Mooshammer et al. (2014) |
| RATIO$_{C:P,stor}$ | C:P ratio of storage | gC/gP | 1200 | Mooshammer et al. (2014) |
| FRAC$_{res,i,l}$ | Daily fraction of plant part $i$ ($i$= root, sapwood,leaf, storage , organ) entering the residues in layer $l$ | d$^{-1}$ | (b) | *cf.* Table A2 |
| RATIO$_{C:N,dec}$ | C:N ratio of soil decomposers | gC/gN | 5 | Mooshammer et al. (2014) |
| RATIO$_{C:P,dec}$ | C:P ratio of soil decomposers | gC/gP | 16 | Mooshammer et al. (2014) |
| VMAX$_{res,soil}$ | Maximum daily remineralization rate in soils | d$^{-1}$ | 0.01 | Schjonning et al. (2004) |
| VMAX$_{immo,X}$ | Maximum daily immobilization rate of nutrient $X$ (N or P) by soil decomposers | d$^{-1}$ | 0.1 | Schjonning et al. (2004) |





| Symbol | Description | Unit | Value | References |
|---|---|---|---|---|
| $m_{dec}$ | Daily mortality of decomposers | $d^{-1}$ | 0.5 | Schjonning et al. (2004) |
| **Phytoplankton growth and remineralization in rivers** | | | | |
| $\mu_{phyto}$ | Maximal daily growth of phytoplankton | $d^{-1}$ | 2.9 | Billen et al. (1994) |
| $K_{N,phyto}$ | Half saturation constant for phyto. growth on N | $gN/m^3$ | 0.07 | Billen et al. (1994) |
| $K_{P,phyto}$ | Half saturation constant for phyto. growth on P | $gP/m^3$ | 0.06 | Billen et al. (1994) |
| $m_{phyto}$ | Daily mortality of phytoplankton and water decomposers, including grazing | $d^{-1}$ | 0.5 | Cebrian (1999) |
| $RATIO_{C:N,phyto}$ | C:N ratio of phytoplankton and water decomposers | gC/gN | 10 | Elser et al. (2000) |
| $RATIO_{C:P,phyto}$ | C:P ratio of phytoplankton and water decomposers | gC/gP | 250 | Elser et al. (2000) |
| sed | Fraction of residues trapped daily into sediments | $d^{-1}$ | 0.024 | Billen et al. (1994) |
| $VMAX_{res,water}$ | Maximum daily remineralization rate in the water | $d^{-1}$ | 0.06 | Billen et al. (1994) |





Table A2: Presentation of the dataset sources used in the simulation.

| Symbol | Description | Unit | Spatial scale | Temporal scale | Dataset sources |
|---|---|---|---|---|---|
| **Wastewater release and treatment** | | | | | |
| $POP_{urban}$ | Population in urban areas | cap | cell | decadal before 2000 yearly after 2000 | HYDE model Klein Goldewijk et al. (2010, 2011) |
| $POP_{rural}$ | Population in rural areas | cap | cell | decadal before 2000 yearly after 2000 | HYDE model Klein Goldewijk et al. (2010, 2011) |
| $FRAC_{sew,urban}$ | Fraction of the urban population that is connected to public sewerage system | - | country | yearly | OECD (2015), The-World-Bank (2016) |
| $FRAC_{sew,rural}$ | Fraction of the rural population that is connected to public sewerage system | - | country | yearly | OECD (2015), The-World-Bank (2016) |
| $FRAC_{sew,i}$ | Fraction of the total population connected to public sewerage system with treatment $i$ (0 for no treatment, 1 for primary/mechanical treatment, 2 for secondary/biological treatment, 3 for tertiary/advanced treatment) | - | country | yearly | OECD (2015) |
| $GDP_{PPP}$ | National per capita gross domestic product based on purchasing power parity | USD1995/prs/yr | country | yearly | The-World-Bank (2016) |
| $GDP_{mer}$ | National per capita gross domestic product based on market exchange rate | USD1995/prs/yr | country | yearly | The-World-Bank (2016) |
| **Nitrification and denitrification in soils and rivers** | | | | | |





| Symbol | Description | Unit | Spatial scale | Temporal scale | Dataset sources |
|---|---|---|---|---|---|
| $T_{soil}$ | Soil temperature | °C | soil | daily | LPJmL model |
| $pH_{soil}$ | Soil pH | | soil | daily | LPJmL model |
| $FRAC_{soil,water}$ | Ratio of soil moisture against soil maximum moisture | | soil layer | daily | LPJmL model |
| $T_{water}$ | Water temperature | °C | cell | daily | LPJmL model |
| **Fertilizer and manure application** | | | | | |
| $FERT_{N,IFA}$ | Amount of N fertilizers applied to soils | gN/y | country | yearly | IFA (2016) |
| $FERT_{P,IFA}$ | Amount of P fertilizers applied to soils | gP/y | country | yearly | IFA (2016) |
| $MAN_{N,pasture}$ | Amount of N manure left on pasture | gN/y | country | yearly | FAO (2016a) |
| $MAN_{N,nopasture}$ | Amount of N manure applied to crops (pastures excluded) | gN/y | country | yearly | FAO (2016b) |
| $N_{NAN}$ | Amount of N manure applied in 1994-2001 | gN/y | country | one year | Potter et al. (2011b) |
| $P_{NAN}$ | Amount of P manure applied in 1994-2001 | gN/y | country | one year | Potter et al. (2011a) |
| **Adsorption in rivers** | | | | | |
| $SURF_{cell}$ | Surface of the cell | km$^2$ | cell | constant | |
| $SURF_i$ | Surface of the system $i$ (forests, grasslands or crops) | km$^2$ | cell | year | LPJmL model |
| $SURF_{urban}$ | Surface of urban areas | km$^2$ | cell | year | HYDE model Klein Goldewijk et al. (2010, 2011) |
| **Phytoplankton growth and remineralization in rivers** | | | | | |
| $VOL_{water}$ | Volume of water | m$^3$ | cell | daily | LPJmL model |
| **Nutrient uptake, remineralization and immobilization in soils** | | | | | |





| Symbol | Description | Unit | Spatial scale | Temporal scale | Dataset sources |
|---|---|---|---|---|---|
| $UPT_{C,i}$ | Daily C uptake for plat part $i$ (root, sapwood, leaf, storage) | g d$^{-1}$ | stand | daily | LPJmL model |
| $FRAC_{root,i}$ | Fraction of total roots by soil layer | - | soil layer | daily | LPJmL model |
| $ROOT_C$ | Total amount of C in roots | gC | stand | daily | LPJmL model |
| $SAP_C$ | Total amount of C in sapwood | gC | stand | daily | LPJmL model |
| $LEAF_C$ | Total amount of C in leaves | gC | stand | daily | LPJmL model |
| $STOR_C$ | Total amount of C in storage | gC | stand | daily | LPJmL model |





Table A3: Local parameters calculated by the LPJmL-Med model.

| Symbol | Description | Unit | Spatial scale | Temporal scale |
|---|---|---|---|---|
| **Wastewater release and treatment** | | | | |
| $SEW_N$ | Daily point sources of N from sewer systems | gN/d | country | daily |
| $SEW_P$ | Daily point sources of P from sewer systems | gP/d | country | daily |
| $WW_{N,urban}$ | Daily per capita N release from waste waters in urban areas | gN/pers/d | country | daily |
| $WW_{P,urban}$ | Daily per capita P release from waste waters in urban areas | gP/pers/d | country | daily |
| $WW_{N,rural}$ | Daily per capita N release from waste waters in rural areas | gN/pers/d | country | daily |
| $WW_{P,rural}$ | Daily per capita P release from waste waters in rural areas | gP/pers/d | country | daily |
| $E_{N,hum}$ | Human N emission | gN/pers/yr | country | yearly |
| $E_{P,hum}$ | Human P emission | gP/pers/yr | country | yearly |
| $E_{P,Ldet}$ | P emission from laundry detergents | gP/pers/yr | country | yearly |
| $E_{P,Ddet}$ | P emission from dishwasher detergents | gP/pers/yr | country | yearly |
| $FRAC_{Pfree,Ldet}$ | Fraction of P-free laundry detergents | - | country | yearly |
| **Nitrification and denitrification in soils and rivers** | | | | |
| $NITR_{soil,l}$ | Daily nitrification in soils | gN d$^{-1}$ | soil layer | daily |
| $DEN_{soil,l}$ | Daily denitrification in soils | gN d$^{-1}$ | soil layer | daily |
| $NITR_{water}$ | Daily nitrification in waters | gN d$^{-1}$ | cell | daily |
| $DEN_{water}$ | Daily denitrification in waters | gN d$^{-1}$ | cell | daily |
| VMAX$_{den,l}$ | Maximum daily denitrification in soil | $d^{-1}$ | soil layer | daily |
| VMAX$_{den,water}$ | Maximum daily denitrification in water | $d^{-1}$ | cell | daily |
| **Fertlizer and manure application** | | | | |
| $FERT_N$ | Daily input of N fertilizers | gN d$^{-1}$ | stand | daily |
| $FERT_P$ | Daily input of P fertilizers | gP d$^{-1}$ | stand | daily |
| $MAN_N$ | Daily input of N manure | gN d$^{-1}$ | per crop per country | daily |
| $MAN_P$ | Daily input of P manure | gP d$^{-1}$ | per crop per country | daily |
| $AREA$ | Area of the crop type considered | ha | stand | yearly |
| $AREATOT$ | Total cultivated area | ha | country | yearly |



| Symbol | Description | Unit | Spatial scale | Temporal scale |
|---|---|---|---|---|
| $AREATOT_{pasture}$ | Total area of pastures | ha | country | yearly |
| $AREATOT_{nopasture}$ | Total cultivated area (pastures excluded) | ha | country | yearly |
| **Irrigation** | | | | |
| $IRR_X$ | Daily input of nutrient $X$ ($N0_3$, $NH_4$ or $PO_4$) through irrigation | g | stand/cell | daily |
| $VOL_{src,irr}$ | Volume of water used for irrigation from source $src$ | $m^3$ | stand/cell | daily |
| **Adsorption in soils and rivers** | | | | |
| $m_l$ | Mass of soil in layer $l$ | g | soil layer | - |
| $VOL_{water,soil}$ | Volume of water in the soil | $m^3$ | soil layer | daily |
| $ADS_{PO4,water}$ | Daily adsorption of phosphates in the water | g | cell | daily |
| $C_{susp,water}$ | Concentration of suspended solids in the water | $g/m^3$ | cell | daily |
| **Nutrient uptake, remineralization and immobilization in soils** | | | | |
| $UPT_X$ | Daily uptake of nutrient $X$ | $g.d^{-1}$ | soil layer | daily |
| $G_{dec,l}$ | Daily consumption of C residues by soil decomp. (layer $l$) | $gC.d^{-1}$ | soil layer | daily |
| $IMMO_N$ | Amount of inorganic N immobilized by soil decomposers | $d^{-1}$ | soil layer | daily |
| $IMMO_P$ | Amount of phosphates immobilized by soil decomposers | $d^{-1}$ | soil layer | daily |
| **Phytoplankton growth and remineralization in rivers** | | | | |
| $G_{phyto}$ | Daily growth rate of phytoplankton | $d^{-1}$ | cell | daily |
| $G_{dec,water}$ | Consumption of carbon residues by water decomposers | $gC.d^{-1}$ | cell | daily |
| **Nutrient leaching and movement** | | | | |
| $NUT_{mobile,l}$ | Concentration of nutrients in the mobile water | $g \cdot m^{-3}$ | cell | daily |
| $W_{mobile}$ | Amount of mobile water in the layer | mm | cell | daily |
| $SAT_l$ | Saturated water content of the soil layer | mm | cell | daily |
| $Q_{surf}$ | Surface runoff (only in the top soil layer) | mm | cell | daily |





| Symbol | Description | Unit | Spatial scale | Temporal scale |
|---|---|---|---|---|
| $Q_{lat,l}$ | Water discharged from the layer by lateral flow | mm | cell | daily |
| $\beta_{nut}$ | Nutrients percolation coefficient | - | - | - |




## Appendix B: The adjustments of dministrative country boundaries:

The international fertilizer association (IFA) provides country-specific yearly N and P fertilizer consumption data since 1961. However, the administrative boundaries of some countries in 1961 do not reflect the 2010 boundaries which are used for running the LPJmL simulations. We face this situation for the five countries Bosnia-Herzegovina, Croatia, Macedonia, Serbia, and Slovenia, which were one single country before 1994: Yugoslavia. Following, before 1994, fertilizer data are provided for Yugoslavia. We use the fertilizer data of the actual countries of the former Yugoslavia between 1995 and 2014 in order to compute the average proportion of fertilizer consumed by each country relative to the total amount consumed by the five countries. We consider the hypothesis that this proportion was the same before 1994 to distribute the fertilizer amounts of the former Yugoslavia between Bosnia-Herzegovina, Croatia, Macedonia, Serbia, and Slovenia for the period 1961-1994.

*Author contributions.* MA, AB, MB, RP, NB, SO contributed to the model development, simulations, and diagnostics. MA, AB, and MB contributed to paper writing and discussions.

*Competing interests.* The authors declare that they have no conflict of interest

*Acknowledgements.* This project is part of the Labex OT-Med (no. ANR-11-LABX-0061) funded by the French Government "Investisse-ments d'Avenir" program of the French National Research Agency (ANR) through the A*MIDEX project (no ANR-11-IDEX-0001-02). This study has been conducted using E.U. Copernicus Marine Service Information. This work has benefited from HPC resources from GENCI-IDRIS (Grant 2020-0227). We acknowledge Anne-Sophie Auguères for her design of the first concept of the land-sea nutrient transfer within LPJmL-Med during her stay at IMBE.





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
