# Peer review of "LPJmL-Med – Modelling the dynamics of the land-sea nutrient transfer over the Mediterranean region–version 1: Model description and evaluation"

_Geoscientific Model Development, 2020_

## Referee Comment (RC1) · Anonymous Referee #1 · 13 Feb 2021

LPJmL_med – Modelling the dynamiscs of the land-sea nutrient transfer over the Mediterranean region – version 1: Model description and evaluation.

This article aims to estimate the discharge as well as nitrate and phosphate export into the Mediterranean Sea with the LPJmL model at 1/12 degree spatial resolution.

General remarks

First of all I realize that the authors put a huge amount of work in the implementation and in writing this lengthy paper. This article tries to document the modeling con-

cept,but this paper does not contain the quality needed for publication. There are a lot of reasons for this (see below). Beside that the structure of the paper could be improved. I had to go back and forth in this paper the whole time. It took me some time to make a review of this article. The formulas presented in this article are not new. The assumptions and the generation should be new. But on this part, I have a lot of questions or remarks. I stopped with remarks on typos and so on, because I believe this article needs a major revision before detailed feedback can be given.

Conceptional remarks

Where is the NH4 in the rivers? Why leave this part of the dissolved N out of this paper? This is needed here as well.

Soil and water temperature are needed in several equations. Where is their description?

Where is the description of the organic part of P and N in the soils?

I miss an overview of the inputs, the delivery to the rivers, the in-stream removal and the export to the sea.

Validation of discharge, NO3 and PO4 should be done on the same time scale. It is very confusing. Why only to the year 2000? That is still 20 years back from now!

Questions

Line 123-124: State variables: N and P contents has a fixed C:N and C:P ratios? Very strange. Why should this be?

Equation 2. Here MAN_x is used, but the description says it is organic matter. So I miss this components in the description of N and P. The organic P storage in the soil behaves totally different than the dissolved P storage. Then equation 4 shows a constant ratio C:P which I really doubt. The storage of P in the soils works different than the storage of C…...

Equation 10: Which part of P fertilizer is available for uptake by plants? Why is the REMIN_PO4 multiplied by PO4. Should that not be organic_P?

Fertilizer only on crop fields. In Europe fertilizer is also used on grassland.

Equation 11: Why is growth of phytoplankton not dependent on NO3 and/or PO4? It turns out that in equation 30 this is not true. Why not moving equation 11 to 2.4.2? I need a lot of searching to find the explanation for some of the processes. Is there another structure of the description possible?

Line 207 – 211: Nice to include this remark, but assumption of N-fixers is not correct, and excluding atmospheric deposition (land and water) is a pitty. So both should be included!

Why is 50% NO3 and 50% NH4 of fertilizer? This is dependent of the type of fertilizer. It is a pitty when you consider NH4 and NO3 and just divide N into two. . .. Some type of fertilizer contains a lot of NH4, some don't. Please improve.

Line 253: What is cultivated land? Cropland and grassland?

Equation 15: Why area weighed distribution and not crop weighed. With LPJmL you have different crop types, so make better use of this information!

Equation 17: Values of Pman and Nman are not given. Why take a constant ratio? It is dependent on the type of animals that produce the manure. So it is country specific and time dependent. . ..

Line 251: Again the authors assume that 50% of sewage effluent is nitrate and 50% is NH4. Why? This is not a reasonable assumption!

Line 255. Where is the input from industry?

I am surprised by taking the waste water model of van Drecht et al. (2009). The problem is the assumptions of the GDPppp and GDPmer which is from US dollars of 1995. There exist almost no model which uses this specific GDPppp and GDPmer. So

the authors will run into problems when applying this model. Besides there are some updates of this model (Moree et al, 2013 and van Puijenbroek et al, 2018). Equations 21 – 24 are dependent on this GDP. . ..

Line 272: I don't think this ratio is the N:P ratio of municipal wastewater. This is the human intake. I miss for a lot of equations the units (for example equations 21 – 24). . .

The C_susp_water is calculated (equation 36). But this is only dependent on landuse. So the amount of suspend matter in the river is independent on the surface runoff or land erosion. Is this assumption valid?

Line 442: So maximal depth of soil is 3 metres. So there is no groundwater modelled here? Also in equations 45 – 49, I don't see any delay in nitrate going through the soil layers. Can you elaborate on this?

Line 512: missing ares?

Line 535: I read that legumes don't get any N and P fertilizer?? Strange.

Line 537: Is there a fertilizer application of grass or not?

Figure 3: typos in headers, What does c and d contain? Load in the main stream of the rivers? Where? At the mouth of the river? What is the unit? In the text it is always PO4-P and NO3-N. But here in the figures? Make clear. How is it possible that PO4 is twice as high as NO3 (c and d)? Why is y-axes of f three times lower than d? The sum of three rivers (c) is higher than sum of two regions. Why? Is a and b fertilizer on the soil? Is NO3 fertilizer in a half of total fertilizer?

Figure 4: I don't understand this figure. Manure is fully organic matter. So NO3 and PO4 are zero. I don't know what is presented here. . ..

Figure 5: I am completely lost. The unit here is in tonnes. Which means that WWT is very very very small compared to fertilizer or manure. Something is wrong. . . . . .

Line 550 – 556: GDP is not used. . ...

Line 566 – 567: "In general . . .. is poorer" Why is this statement here?

Figure 6: Why is the Nile here? There is no comparison with observations. Why upper figure average of 1963 – 2000? Figure caption claims 1920 – 1985, that is only for Ebro.

Line 576: Why limit the impact of damming and anthropogenic water use? This is one of the things your model can do, so I don't understand this. Why mentioning "but not Nile"?

I think there is a misunderstanding in Figure 7. Here there is a comparison between modeled data and observed data. I miss the 1:1 line. What calibration data is used to calibrate the discharge of this model? At which location is this comparison? Why here 1920 – 1980 and not 1985?

Line 586: Why the word "may"? Do you mean that Lutz and/or Ludwig is claiming this or are you seeing this in the model data?

Line 590: the role of dams. Why figure 8? Is it a claim of this model also to reproduce the monthly discharge? Why not show the whole timeserie? I would not include this figure.

Line 613: the word "could"? Are you not sure?

Figure 9: There is something wrong with the model. The trend of observed NO3 is not reproduced by the model. This is a problem, because it is becoming worse when the time is closer to the current situation. The same holds for Figure 10 for PO4. Same questions for these two figures? Where in the rivers are these comparisons?

In conclusions the word "concentration" is used. I did not see any concentrations in this article.

---

## Short Comment (SC1) · 26 Feb 2021

Dear authors,

in my role as Executive editor of GMD, I would like to bring to your attention our Editorial version 1.2:

https://www.geosci-model-dev.net/12/2215/2019/

This highlights some requirements of papers published in GMD, which is also available

on the GMD website in the 'Manuscript Types' section: http://www.geoscientific-model-development.net/submission/manuscript_types.html

In particular, please note that for your paper, the following requirement has not been met in the Discussions paper:

- Code must be published on a persistent public archive with a unique identifier for the exact model version described in the paper or uploaded to the supplement, unless this is impossible for reasons beyond the control of authors. All papers must include a section, at the end of the paper, entitled "Code availability". Here, either instructions for obtaining the code, or the reasons why the code is not available should be clearly stated. It is preferred for the code to be uploaded as a supplement or to be made available at a data repository with an associated DOI (digital object identifier) for the exact model version described in the paper. Alternatively, for established models, there may be an existing means of accessing the code through a particular system. In this case, there must exist a means of permanently accessing the precise model version described in the paper. In some cases, authors may prefer to put models on their own website, or to act as a point of contact for obtaining the code. Given the impermanence of websites and email addresses, this is not encouraged, and authors should consider improving the availability with a more permanent arrangement. Making code available through personal websites or via email contact to the authors is not sufficient. After the paper is accepted the model archive should be updated to include a link to the GMD paper.

As GitLab is not a persistent archive, please provide a persistent release for the exact source code version used for the publication in this paper. As explained in https://www.geoscientific-model-development.net/about/manuscript_types.html the preferred reference to this release is through the use of a DOI which then can be cited in the paper. For projects in GitHub a DOI for a released code version can easily be

created using Zenodo, see https://guides.github.com/activities/citable-code/ for details. Finally note, that according to our new Editorial (v1.2) all data and analysis / plotting scripts should be made available.

Yours, Astrid Kerkweg

―――――――――――――――――

---

## Referee Comment (RC2) · Anonymous Referee #2 · 15 Jun 2021

The paper modeled the dynamics of water discharge, nitrate and phosphorus export to the Mediterranean Sea with the LPJmL model. The authors integrated a number of formulas in quantifying processes of N and P losses into rivers, in-river retention, and export. These processes include remineralization, adsorption, nitrification, denitrification and phytoplankton dynamics, etc. Indeed, these formulas in quantify the processes are not new, but the model runs on daily time scale. The key question is modeling N and P input into rivers from terrestrial sources with effective validation, which would

make the model robust. That is particularly difficult and challenging. Up to now, less study or modeling can be validated in this area.

From technical point of view, I have following questions regarding N and P input into rivers, in-river and in-reservoir retentions. Figure 1, regarding the conceptual diagram of nitrate and phosphate losses to rivers, diffuse sources include soil source, urban land source, rural land source. In addition, aquacultures feed also should be considered. This paper only modeled the soil diffuse source, without quantifying other diffuse sources such as urban land source, rural land source and aquacultures feed. For soil diffuse source, I concern the impact of cropping system change on N and P budgets, this is particularly important from long-term agricultural activities. I am particularly interested in in-river and in-reservoir retention of N and P. For in-reservoir retention, the temporal pattern in construction of artificial dam-reservoirs can significantly influence reservoir's retention/removal of N and P. A group of newly-built reservoirs in upriver can influence downriver "aged" reservoir's retention of N and P, particularly for P, because the inputs of N and P to these "aged" reservoirs have been changed due to the newly-built reservoirs in the same river networks. For in-river retention, N and P retention by different river orders should be considered at the river network scale, because rivers with different orders have different hydraulic loads which can control N and P retention. Finally, the model runs on daily time scale, authors should showed the daily-temporal changes in nitrate and phosphate concentrations for those rivers flowing into the Mediterranean Sea.

---

## Author Comment (AC1) · 25 Jun 2021

**Response to the comments about the submitted paper "LPJmL-Med - Modelling the dynamics of the land-sea nutrient transfer over the Mediterranean region-version 1: Model description and evaluation"**

**Mohamed Ayache, Alberte Bondeau, Rémi Pagès, Nicolas Barrier, Sebastian Ostberg, and Melika Baklouti**

We would like to thank Reviewer 1 for taking the time and effort necessary to review the manuscript. We sincerely appreciate all valuable comments and suggestions, which helped us to improve the quality of the manuscript (ms hereafter).

Please note that reviewers' comments are in blue while our answers are in black.

**Reviewer R1**

First of all I realize that the authors put a huge amount of work in the implementation and in writing this lengthy paper. This article tries to document the modeling concept, but this paper does not contain the quality needed for publication. There are a lot of reasons for this (see below). Beside that the structure of the paper could be improved. I had to go back and forth in this paper the whole time. It took me some time to make a review of this article. The formulas presented in this article are not new. The assumptions and the generation should be new. But on this part, I have a lot of questions or remarks. I stopped with remarks on typos and so on, because I believe this article needs a major revision before detailed feedback can be given.

We would like to thank the reviewer for highlighting the main points that should be considered under revision. The reviewer is right, this paper is the fruit of a considerable effort, starting from the implementation of the biogeochemical land-sea nutrient transfer processes within the LPJmL model, followed by the preparation of the inputs/boundary conditions of the model which required the manipulation of a large number of database, the calibration/evaluation of the model, and finally the writing of the paper.

We are aware that our approach still suffers from a lot of limitations and that the structure of the paper could be improved, but it intends to be a first proof of concept of the modelling of the transfer of nutrients from land to oceans. We hope that the corrections we brought based on the relevant advice from the reviewer help to clarify and improve the present paper.

Where is the NH4 in the rivers? Why leave this part of the dissolved N out of this paper? This is needed here as well.

As shown in the paper (section 2), NH4 is also implemented in LPJmL-Med represented through its dynamics in soil (equation 9) and in water (equation 13). Transformations between different forms of N in the soil are represented by remineralization, nitrification, and denitrification and are simulated in sequential order. We did not show the results of NH4, because in-situ data covering the whole Mediterranean basin are not available, which limits the evaluation of NH4 output, and on the other hand, the Mediterranean Sea is an exception with a strong P limitation (where N is generally considered to limit primary productivity in most of the world's oceans). For these reasons, and because of the greater availability of data on NO3 and PO4, we decided to focus hereafter on these two macro-nutrients. However, as suggested by the reviewer, a figure of NH4 results has been added in the appendix of the paper (see the Fig. R1 above, see Figure A1 in the revised manuscript). In addition, a sentence has been added to the text to make this point perfectly clear in the revised manuscript (see sections 1 and 5.2.1).

---

## Author Comment (AC2) · 24 Jul 2021

**Response to the comments about the submitted paper "LPJmL-Med – Modelling the dynamics of the land-sea nutrient transfer over the Mediterranean region–version 1: Model description and evaluation"**

**Mohamed Ayache, Alberte Bondeau, Rémi Pagès, Nicolas Barrier, Sebastian Ostberg, and Melika Baklouti**

We would like to thank Reviewer 1 for taking the time and effort necessary to review the manuscript. We sincerely appreciate all valuable comments and suggestions, which helped us to improve the quality of the manuscript (ms hereafter).

Please note that reviewers' comments are in blue while our answers are in black.

Reviewer R2

The paper modeled the dynamics of water discharge, nitrate and phosphorus export to the Mediterranean Sea with the LPJmL model. The authors integrated a number of formulas in quantifying processes of N and P losses into rivers, in-river retention, and export. These processes include remineralization, adsorption, nitrification, denitrification and phytoplankton dynamics, etc. Indeed, these formulas in quantify the processes are not new, but the model runs on daily time scale. The key question is modeling N and P input into rivers from terrestrial sources with effective validation, which would make the model robust. That is particularly difficult and challenging. Up to now, less study or modeling can be validated in this area.

We would like to thank the reviewer for this comment. We agree withe the referee that the modeling of N and P input into rivers from terrestrial sources is particularly difficult and challenging for agro-ecosystem model. This paper is the fruit of a considerable effort on the implementation of the biogeochemical land-sea nutrient transfer processes and the preparation of the inputs/boundary conditions of the model.

From technical point of view, I have following questions regarding N and P input into rivers, in-river and in-reservoir retentions. Figure 1, regarding the conceptual diagram of nitrate and phosphate losses to rivers, diffuse sources include soil source, urban land source, rural land source. In addition, aquacultures feed also should be considered.This paper only modeled the soil diffuse source, without quantifying other diffuse sources such as urban land source, rural land source and aquacultures feed.

We are not sure what does the reviewer mean by "diffuse urban and rural land source". This paper considers both point sources and diffuse sources. Point sources deliver nutrients through wastewater release, this occurs at any location with population, whatever urban or rural. Could such "point sources" be what does the reviewer mean by "urban and rural point sources", and being considered as "diffuse" because population is distributed in every cell? Usually what we state as "diffuse sources" are the agriculture sources, which deliver nutrients to the soil. Through leaching, they percolate with water towards rivers. We agree with the reviewer that onshore aquaculture should be considered, as this sector contributes to nutrient load in the Mediterranean [Gorjanc et al., 2020] despite many programs for treating or recycling this load. Several studies on fresh water quality and modelling exercises which account for the impact of fish farming can be

found in the literature but they are mostly local [Kontogianni et al., 2007, Kagalou et al., 2012] and focus often on the coastal zone. The integration of such nutrient load source within the whole Mediterranean catchment would need a huge data collection (spatio-temporal dynamics of aquaculture ponds, their typology for estimating regional mean load and cleaning processings, policy regulation time-series, etc. Such a work could not be done within the present study, it should be the objective of another paper. We have explained this point within the ms, both in the introduction and in the discussion.

However, we are aware that our approach still suffers from a lot of limitations, and the implementation of the aquacultures feed input in the model could be among the future improvement of the model.

For soil diffuse source, I concern the impact of cropping system change on N and P budgets, this is particularly important from long-term agricultural activities. I am particularly interested in in-river and in-reservoir retention of N and P. For in-reservoir retention, the temporal pattern in construction of artificial dam-reservoirs can significantly influence reservoir's retention/removal of N and P. A group of newly-built reservoirs in upriver can influence downriver "aged" reservoir's retention of N and P, particularly for P, because the inputs of N and P to these "aged" reservoirs have been changed due to the newly-built reservoirs in the same river networks.

The cropping system is a very important driver of N and P budgets, so any change in the system will impact those budgets. The land use input data for LPJmL have been compiled from the combination of different sources, among other the decadal cropland and grassland data from the HYDE3.1 dataset [Goldewijk et al., 2011], which allow us to account for the land use change dynamics. We have constructed an input data set with yearly crop and managed grassland distribution data since the year 1700. This is explained in section 4.1.1 of the ms, that we have slightly modified in order to make clearer the dynamic properties of the land use input data. (cf. 1st paragraph of section 4.1.1 in the revised manuscript)

*The land use data for the crops in LPJmL-Med had been compiled from different sources, as explained in [Fader et al., 2015, Fader et al., 2010]. Decadal cropland and managed grassland data from HYDE3.1 [Goldewijk et al., 2011] were interpolated to derive annual values and then used for extrapolating the detailed crop distribution patterns of 2000 ([Portmann et al., 2010, Monfreda et al., 2008]) to the past, until 1700. Historical irrigation fractions were determined as explained in Fader et al. (2010). Further information is given by [Fader et al., 2015, Fader et al., 2010]. Following, this input dataset accounts for both land use change since 1700 and one important indicator of agriculture intensification, i.e. irrigation.*

For in-river retention, N and P retention by different river orders should be considered at the river network scale, because rivers with different orders have different hydraulic loads which can control N and P retention.

Denitrification of $NO_3$ and adsorption of $PO_4$ are the main processes leading to N and P retention in rivers (see e.g. [Billen and Garnier, 2007]).

We compute the $PO_4$ adsorption in rivers following [Billen and Garnier, 2007], where $PO_4$ adsorption depends on the water volume and on the concentration of suspended solids, which we compute as a function of the land use, as explained in section 2.4.3. In such formulation, possible

hydraulic load effect on P retention is not accounted for.

On the other hand, we consider the impact of hydraulic load on N retention (see equation 44), process reported in the literature since several decades when statistical analyses have shown that higher annual hydraulic loads are correlated with lower nutrient retention (see e.g. [Behrendt and Opitz, 1999]). However, we acknowledge that we had to improve the computation of the hydraulic load, which was a simple constant value in our equation. We now redo the computation of the hydraulic load by considering the seasonal dynamic of the water discharge and the river dimensions (this shows differences implicitly related to the river orders). We also take into account the seasonal temperature variations in the computation of the uptake velocity for denitrification.

Finally, the model runs on daily time scale, authors should showed the dailytemporal changes in nitrate and phosphate concentrations for those rivers flowing into the Mediterranean Sea.

In deed the model runs on daily time scale, but it is forced by monthly climate inputs data that are daily interpolated (considering the distribution of the monthly precipitation on the number of rainy days randomly distributed). See section 4.1 describing the model inputs. Following the simulated daily nitrate and phosphate concentrations cannot be considered to be realistic, when the monthly values are. Furthermore, the marine model that will use these simulated nitrate and phosphate concentrations (in another exercise) runs with monthly nutrient forcing inputs. The LPJmL simulations at a monthly time scale are therefore well adapted to such a modem cascade, as this is indicated in the discussion.

**References**

[Behrendt and Opitz, 1999] Behrendt, H. and Opitz, D. (1999). Retention of nutrients in river systems: dependence on specific runoff and hydraulic load. *Hydrobiologia 1999 410:0*, 410:111–122.

[Billen and Garnier, 2007] Billen, G. and Garnier, J. (2007). River basin nutrient delivery to the coastal sea: Assessing its potential to sustain new production of non-siliceous algae. *Marine Chemistry*, 106:148–160.

[Fader et al., 2015] Fader, M., Bloh, W. V., Shi, S., Bondeau, A., and Cramer, W. (2015). Modelling mediterranean agro-ecosystems by including agricultural trees in the lpjml model. *Geoscientific Model Development*, 8:3545–3561.

[Fader et al., 2010] Fader, M., Rost, S., Müller, C., Bondeau, A., and Gerten, D. (2010). Virtual water content of temperate cereals and maize: Present and potential future patterns. *Journal of Hydrology*, 384:218–231.

[Goldewijk et al., 2011] Goldewijk, K. K., Beusen, A., Drecht, G. V., and Vos, M. D. (2011). The hyde 3.1 spatially explicit database of human-induced global land-use change over the past 12,000 years. *Global Ecology and Biogeography*, 20:73–86.

[Gorjanc et al., 2020] Gorjanc, S., Klančnik, K., Murillas-Maza, A., Uyarra, M. C., Papadopoulou, N. K., Paramana, T., Smith, C., Chalkiadaki, O., Dassenakis, M., and Peterlin, M. (2020). Coordination of pollution-related msfd measures in the mediterranean - where we stand now and insights for the future. *Marine Pollution Bulletin*, 159:111476.

[Kagalou et al., 2012] Kagalou, I., Leonardos, I., Anastasiadou, C., and Neofytou, C. (2012). The dpsir approach for an integrated river management framework. a preliminary application on a mediterranean site (kalamas river -nw greece). *Water Resources Management 2012 26:6*, 26:1677–1692.

[Kontogianni et al., 2007] Kontogianni, A., Kavakli, Z., Avagianou, E., and Tsirtsis, G. (2007). Implementation of integrated scenario analysis and modelling for the sustainable development of a coastal area in eastern mediterranean. *Transitional Waters Monographs*, 1:93–105.

[Monfreda et al., 2008] Monfreda, C., Ramankutty, N., and Foley, J. A. (2008). Farming the planet: 2. geographic distribution of crop areas, yields, physiological types, and net primary production in the year 2000. *Global Biogeochemical Cycles*, 22.

[Portmann et al., 2010] Portmann, F. T., Siebert, S., and Döll, P. (2010). Mirca2000-global monthly irrigated and rainfed crop areas around the year 2000: A new high-resolution data set for agricultural and hydrological modeling. *Global Biogeochemical Cycles*, 24:n/a–n/a.

---

## Author Comment (AC3) · 24 Jul 2021

Dear Astrid Kerkweg,

Thank you so much for your detailed explanation on the publication of model code and data with a digital object identifier.

According to your recommendation, we have made a DOI to our code using Zenodo archiving tool, by crating a new DOI for our GitHub repository.

The DOI of our code is:

10.5281/zenodo.5133728

Available here:

https://doi.org/10.5281/zenodo.5133728

We will added this DOI in the revised manuscript

Best regards Mohamed Ayache